# Intelectin-2 is a broad-spectrum antimicrobial lectin

**Amanda E. Dugan[1,9], Deepsing Syangtan** [1,9]**, Eric B. Nonnecke** [2]**, Rajeev S. Chorghade[1], Amanda L. Peiffer** [1]**, Jenny J. Yao[1], Jessica Ille-Bunn[1], Dallis Sergio** [3]**, Gleb Pishchany[3], Catherine Dhennezel** [3]**, Hera Vlamakis[3], Sunhee Bae[1], Sheila Johnson[4], Chariesse Ellis[4], Soumi Ghosh[5], Jill W. Alty[1], Carolyn E. Barnes[1], Miri Krupkin** [6]**, Gerardo Cárcamo-Oyarce** [6]**, Katharina Ribbeck** [6]**, Ramnik J. Xavier** [3,7]**, Charles L. Bevins** [2] **& Laura L. Kiessling** [1,3,4,8] ✉

Mammals regulate the localization, composition, and activity of their native microbiota at colonization sites. Lectins residing at these sites influence microbial populations, but their functional roles are often unclear. Intelectins are found in chordates at mucosal barriers, but their functions are not well characterized. In this study, we find that mouse intelectin-2 (mItln2) and human intelectin-2 (hItln2) engage and crosslink mucins via carbohydrate recognition. Moreover, both lectins recognize microbes within native microbial communities, including gram-positive and gram-negative isolates from the respiratory and gastrointestinal tracts. This ability to engage mammalian and microbial glycans arises from calcium-coordinated binding of carbohydrate residues within mucus and microbial surfaces. Microbes, but not human cells, bound by mItln2 or hItln2, suffer a loss of viability. These findings underscore the crucial antimicrobial role of mammalian intelectin-2 in mucosal defense, where it plays offensive (microbial killing) and defensive (mucus crosslinking) roles in regulating microbial colonization.

Mucosal surfaces of the mammalian respiratory and gastrointestinal tracts are replete with microorganisms seeking to colonize these host tissues. Under homeostatic conditions, microbial communities flourish and contribute to host health. Simultaneously, exposure to opportunistic pathogens, allergens, and helminths can injure host tissues and lead to infection. Host tissues must therefore respond to an ever-changing landscape of immunological challenges. In response to microbial colonization, host cells secrete a suite of soluble factors that regulate the localization, composition, and activity of microbial communities. Among these factors are lectins, carbohydrate-binding proteins that bind glycans on host and microbial cells[1–4]. The intelectins, or X-type lectins, are secreted at mucosal surfaces and, though purported to play a role in host defense against microbes, aspects of their function are still unclear[5–11]. Human and mouse intelectin-1 (Itln1) are constitutively expressed in the intestines and exclusively recognize a set of microbial glycans[12–16]. Itln1 has been suggested to facilitate phagocytic uptake of microbes by neutrophils and regulate the localization of mucolytic microbes[8,10,13]. In addition to Itln1, humans and certain mouse strains possess a second intelectin gene, intelectin-2

[1]Department of Chemistry, Massachusetts Institute of Technology, Cambridge, MA, USA. [2]Department of Microbiology and Immunology, School of Medicine, University of California-Davis, Davis, CA, USA. [3]The Broad Institute of MIT and Harvard, Cambridge, MA, USA. [4]Department of Biochemistry, University of Wisconsin-Madison, Madison, WI, USA. [5]Department of Biology, Massachusetts Institute of Technology, Cambridge, MA, USA. [6]Department of Biological Engineering, Massachusetts Institute of Technology, Cambridge, MA, USA. [7]Department of Molecular Biology, Massachusetts General Hospital and Harvard Medical School, Boston, MA, USA. [8]Koch Institute for Integrative Cancer Research, Massachusetts Institute of Technology, Cambridge, MA, USA. [9]These authors contributed equally: Amanda E. Dugan, Deepsing Syangtan. ✉e-mail: kiesslin@mit.edu

(Itln2), which has poorly characterized ligand specificity and unclear biological function[6,17–20]. Mouse intelectin-2 (mItln2) has been reported to be upregulated during nematode infections. Human intelectin-2 (ITLN2, henceforth hItln2) is constitutively expressed and restricted to the small intestine. Moreover, in the small intestine of ileal Crohn's disease (CD) patients, the expression of hItln2 is decreased while it is increased in the colonic tissue of patients with colonic CD and ulcerative colitis (UC). Given that intelectin-2 is implicated in inflammatory processes and disorders involving microbial dysbiosis, we investigated its ligand specificity and function[15,21–26].

Herein, we report the characterization of mItln2 and hItln2. Using a novel mouse model with an intact intelectin locus, we found that mItln2 is selectively and potently induced by T-helper type 2 (Th2) stimuli. We show that mItln2 and hItln2 are indeed lectins, as they recognize select glycans with terminal β-D-galactopyranose residues, which are found on mammalian and microbial glycans. Regarding the host, these lectins cross-link mucins in a glycan-dependent manner. Regarding microbes, these lectins bind a range of gram-positive and gram-negative bacteria. Moreover, we found that mItln2 exerts direct microbicidal effects. HItln2 binding also limits microbial growth, in this case through agglutination, and its activity is intriguingly more potent at lower pH and salt conditions. Taken together, our findings reveal the roles of Itln2 in host defense —through mucin cross-linking and broad-spectrum antimicrobial activity—underscoring the multifaceted roles of intelectins in host–microbe interactions.

## Results

### MItln2 expression is induced by IL-4 and IL-13 in enteroids

In Th2 inflammation, microbial-, helminth-, and allergen-derived toxins and proteases compromise the integrity of host mucosal barriers[27–29]. Such damage can increase the risk of infection by opportunistic pathobionts and prompt the host to activate wound repair and defense pathways. There is evidence that mammalian intelectins can be induced during Th2-type immune responses, including during intestinal helminth infections and asthma, but many details are unclear[6,7,17,20,23,24,30].

We used a mouse model to probe the regulation of intelectin expression during inflammation. The widely used strains for studying inflammation, such as C57BL/6 and its sub-strains (e.g., C57BL/10), encode a single intelectin gene, *Itln1*, resulting from a large 420-kb deletion across the intelectin locus[9,17,18,31,32]. However, most wild-derived and conventional laboratory mouse strains encode six intelectin paralogs: *Itln1, Itln2, Itln3, Itln4, Itln5,* and *Itln6*[18,31]. To better understand intelectin regulation, we generated a congenic C57BL/6 mouse model (B6.C-*Itln1-6*) that encodes a complete *Itln1-6* locus derived from the BALB/c strain[18,33]. Stimulation of enteroids from B6.C-*Itln1-6* mice with type 2 cytokines (IL-4 and IL-13) resulted in a six-fold increase in total *Itln* expression compared to a two-fold increase in wild-type C57BL/6 enteroids (Fig. 1A, B). In tissue and unstimulated enteroids from B6.C-*Itln1-6* mice, most (≥94%) of total intelectin mRNA was *Itln1*, with 2–5% being *Itln2* (Fig. 1C, D). Following IL-4 and IL-13 treatment, the relative abundance of *Itln2* increased to 30% of total transcript reads, indicating a 40-fold upregulation (Fig. 1D;

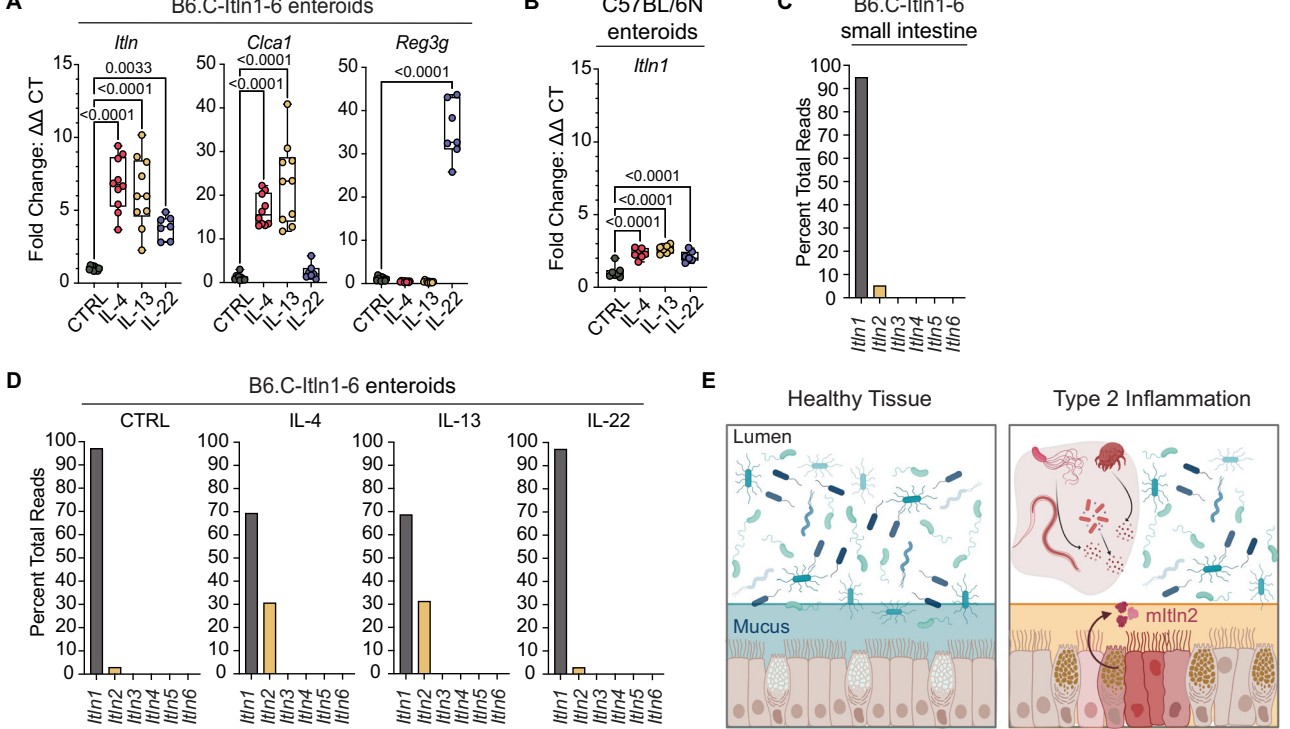

**Fig. 1 | Mouse intelectin-2 (mItln2) is conditionally expressed in type 2 inflammation. A** Relative quantification (mean ± SEM) of total *Itln* expression in enteroids derived from B6.C-*Itln1-6* mice after treatment with different cytokines. Goblet cell product *Clca1* and intestinal epithelial cell product *Reg3g* served as positive controls for IL-4/ IL-13 and IL-22 treatments, respectively (collective data from 3 independent experiments). **B** Relative quantification (mean ± SEM) of *Itln1* expression in enteroids derived from C57BL/6 N mice after treatment with different cytokines (collective data from 3 independent experiments). (**C, D**) Percentage of total *Itln* sequencing reads mapped to different *Itln* paralogs. Data represent percent reads from bulk RNAseq analysis of pooled samples from distal small intestine in B6.C-*Itln1-6* mice (n = 3) (**C**) and B6.C-*Itln1-6*-derived enteroids stimulated with different cytokines (n = 6 independent experiments) (**D**). **E** Schematic of mItln2 expression in mouse lungs and intestines upon exposure to helminths and allergens. Created in BioRender. Lab01, K. (2025) https://BioRender.com/6ylynqc. Statistical analysis in (**A**) and (**B**) involves one-way ANOVA with Dunnett's multiple comparisons test. Box and whisker plots show all data points with center line representing the median, the box is the interquartile range (Q1–Q3), and whiskers extend to absolute maximum and minimum values in the dataset. For complete data, see the source data file.

Supplementary Table 1). This finding aligns with previous report that *Itln2* is robustly induced in the small intestine of BALB/c mice infected with *Nippostrongylus brasiliensis*, *Trichuris muris*, and *Trichinella spiralis*, organisms that promote Th2 inflammation[6,17,20]. As Stat6 is critical for Th2 inflammation, we examined the promoter of *mItln2*. In the BALB/c mouse genome, the *mItln2* promoter region (−2000 base pairs to −1 base pair upstream of transcription start site) has a putative Stat6-binding motif, further supporting our observation of Th2-dependent expression of mItln2 (Supplementary Fig. 1A)[34]. No other intelectins (*Itln3-6*) were detected in small intestinal tissue or enteroids at baseline or after type 2 cytokine stimulation. Type-1 associated cytokines were not evaluated in the enteroids as these cytokines, including interferon gamma, can affect the viability of goblet and Paneth cells[35,36]. Instead, we selected the type 17/22 inflammatory cytokine, IL-22, as a control because IL-22 can stimulate the production of antimicrobial proteins, including Reg family proteins[37]. In these assays, IL-22 showed little or no effect on mItln2 expression. These results establish that Th2-type cytokines selectively induce mItln2 (Fig. 1E). These data suggest a role for mItln2 distinct from that of other intelectins.

## MItln2 recognizes select glycans with terminal β-D-galactopyranose residues

Mouse Itln2 shares greater than 80% sequence identity with the homologous protein Itln1 (Supplementary Fig. 1B). This similarity prompted us to analyze the structure and function of mItln2. To this end, we generated a recombinant form of mItln2 containing a StrepII tag that facilitated purification and detection (Supplementary Fig. 1C, D). We found that strep-mItln2 was glycosylated and properly folded with a melting temperature of approximately 53 °C (Supplementary Fig. 1E–I; Supplementary Fig. 2A). Circular dichroism analysis further revealed that mItln2 exhibits a mixed alpha-helical and beta-sheet secondary structure, suggesting a comparable folded structure as Itln1 (Supplementary Fig. 1E). Although mItln2 lacks the two cysteines present in hItln1 that form a disulfide-linked homotrimer (Supplementary Fig. 1B)[12,34], treatment of mItln2 with the bifunctional homo-cross-linker bis-sulfosuccinimidyl suberate gave rise to a laddered mixture of monomer, dimer, trimer, and hexamer species, suggesting non-covalent oligomerization of mItln2 (Supplementary Fig. 2B). Dynamic light scattering also showed that mItln2 occupies a variable distribution of oligomeric species in solution (Supplementary Fig. 2C). Similarly, the structural prediction of mItln2 trimer using AlphaFold 3 yielded highly confident structures in which the ligand-binding domain from each monomer is facing the same side, as in the previously resolved hItln1 structures (Supplementary Fig. 2d)[2,38]. Together, these data indicate that mItln2 is a stably folded, glycosylated lectin that assembles into non-covalent oligomers. Additionally, the observed laddered oligomerization is consistent with other lectins that form higher-order quaternary structures to achieve avid binding to their cognate ligands[1,2].

Human and mouse Itln1 were previously shown to have identical binding sites and bind exclusively to microbial carbohydrates, including β-D-galactofuranose (β-Galf), the five-membered ring isomer of galactose[12,14]. Despite the high sequence similarities of mItln2 with mItln1 and hItln1, the ligand-binding domain of mItln2 differs in a single residue (W288A) (Fig. 2A; Supplementary Fig. 1B). The aromatic box formed by W288 and Y297 is crucial for Itln1's specificity for microbial monosaccharides. The parsimonious binding site change between mItln1 and mItln2 suggests that the latter might exhibit different ligand specificities than mItln1 and hItln1, leading to differences in microbial binding[13].

To assess whether mItln2 is a glycan-binding protein, we screened the recombinant protein on multiple mammalian and microbial glycan arrays to ensure a robust analysis (Supplementary Fig. 3A). Array hits and intensities cannot be directly compared because of differences in their ligand composition, surface-immobilization chemistries, and

imaging instrumentation; however, all arrays revealed hits containing terminal galactopyranose residues. Microbial glycan analysis yielded hits including yeast and plant-like galactomannan (Davanat) and bacterial galactose-containing capsular polysaccharides (Fig. 2B, C; Supplementary Data 1)[39]. In addition, mItln2 bound mammalian glycans on the National Consortium for Functional Glycomics (NCFG) and commercial glycan arrays, including several oligosaccharides from N- and O-glycans displaying terminal α- or β-D-galactopyranose or 6-SO₃-galactopyranose (Fig. 2B, C; Supplementary Fig. 3B, C; Supplementary Data 1). Moreover, the NCFG glycan array showed binding primarily to bi- and tri-antennary ligands with three or more repeating *N*-acetyllactosamine (LacNAc) units, which are composed of β-D-galactopyranose (β-Galp) and β-D-*N*-acetyl-glucosamine (β-GlcNAc). While the commercial mammalian glycan array also showed mItln2 binding to terminal Galp, there was greater hit diversity regarding the extent of branching, number of LacNAc repeats, and identity of terminal Galp-containing disaccharides (e.g., Galβ1-3GalNAc). Notably, the binding profile of mItln2 had no overlap with hItln1, indicating these related lectins had distinct specificity. We previously reported that hItln1, like mItln1, binds to acyclic vicinal (1,2)-diol moieties present on microbial but not mammalian glycans (Supplementary Fig. 3D)[2,12].

A limitation of glycan array data is its lack of information on relative affinities or ranked binding preferences. The apparent binding signal intensities within an array may reflect differences in glycan density, presentation context, or other technical factors rather than true affinity differences. We therefore validated the glycan array results through complementary biochemical approaches. To quantify the array results and assess mItln2 minimal binding groups, we used an enzyme-linked lectin assay (ELLA). In this assay, biotinylated carbohydrates were immobilized on streptavidin-coated plates and increasing concentrations of mItln2 were added. Plates were washed to remove unbound lectin, and bound mItln2 was detected with anti-StrepII-HRP antibody (Supplementary Fig. 3E, F). Consistent with the array results, these experiments demonstrated that mItln2 binds β-*N*-acetyllactosamine (β-LacNAc), and β-6-SO₃-*N*-acetyllactosamine (β-6SO₃ LacNAc) (Supplementary Fig. 3G). Because β-LacNAc is a disaccharide composed of terminal β-Galp linked to β-GlcNAc, we also evaluated binding to individual monosaccharides to determine mItln2 specificity. ELLA results show that mItln2 does not bind β-GlcNAc or α-Galp and only recognizes β-Galp (Supplementary Fig. 3G). The lectin binding was dose-dependent, a hallmark of specific recognition. Additional experiments using biolayer interferometry (BLI) indicate that mItln2 selectively binds the pyranose form of galactose (β-Galp) but not the furanose form (β-Galf) recognized by Itln1 (Supplementary Fig. 3H–J). The apparent dissociation constant of mItln2 for β-Galp is approximately 1.5 μM (Fig. 2D; Supplementary Fig. 3K). Because mItln2 appears to accommodate both β-Galp and β-6SO₃-Galp as ligands, we postulate that it uses calcium to coordinate binding to conserved hydroxyl groups on the pyranose ring of galactose, a feature shared by all glycan hits. Still, mItln2 does not bind every β−Galp-containing glycan on the array, indicating there may be additional subsites that contribute to affinity. The prevalence of bi- and tri-antennary ligands as top hits on the arrays also suggests that this kind of glycan may have favorable orientation or display for oligomeric mItln2 (Supplementary Fig. 2B, D). Notably, a single amino acid difference in the binding site of mItln2 allows this lectin to preferentially bind the pyranose form of galactose over the furanose form preferred by Itln1.

## MItln2 cross-links mucin glycoproteins

During parasitic nematode infections, mucins are secreted by goblet cells into the intestinal and lung mucosa, where they can interact with newly expressed mItln2[17]. Mucins are glycosylated with epitopes similar to those of the top mammalian glycan array hits for mItln2; therefore, we assayed mItln2 interactions with gastric and intestinal

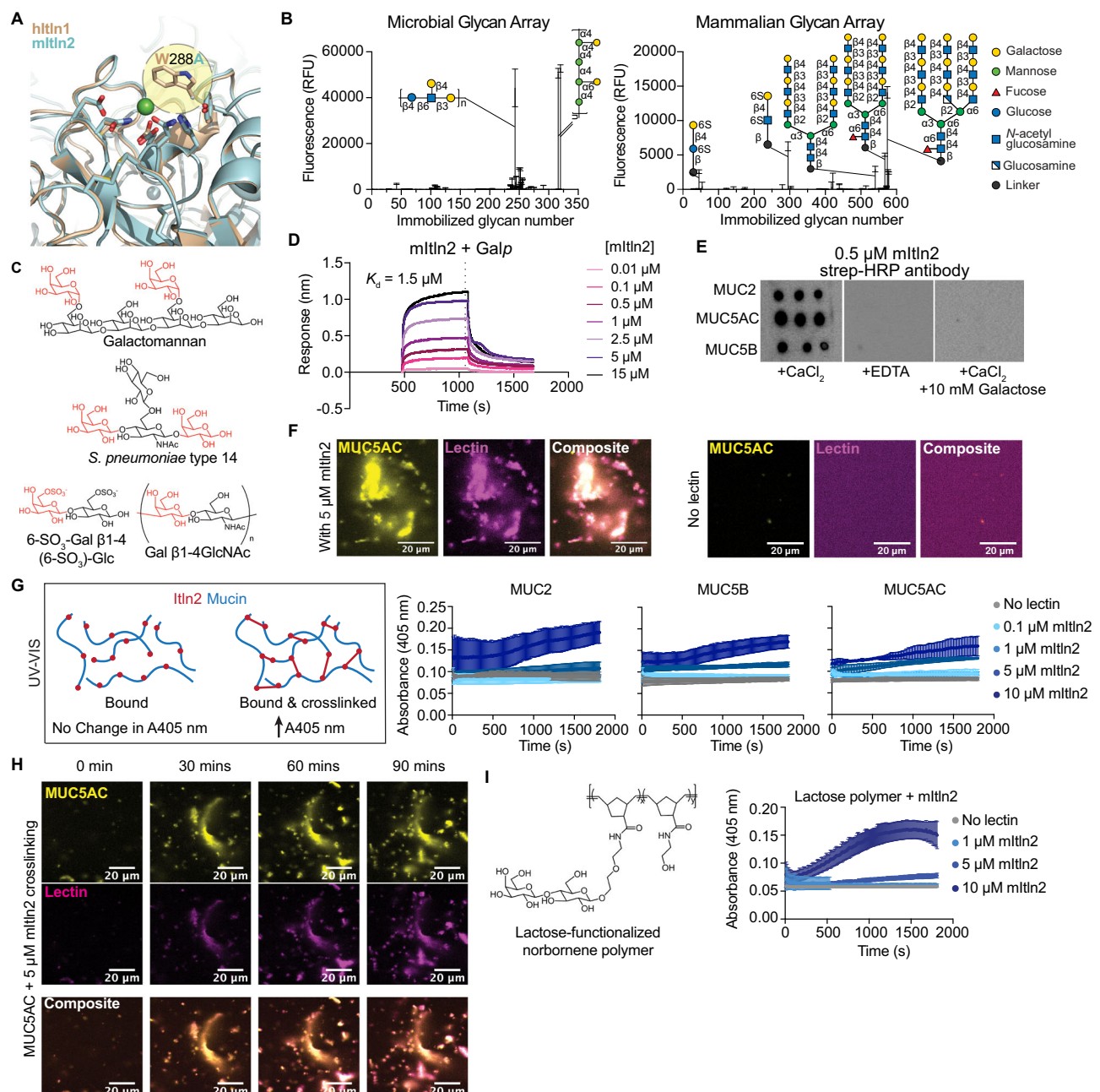

**Fig. 2 | Characterization of glycan specificity for mItln2. A** Overlay of carbohydrate recognition domain of hItln1 (wheat, PDB ID 4WMY) and mItln2 (teal, predicted model), showing calcium (green), conserved calcium coordination residues, and the presence of alanine instead of tryptophan at position 288 in mItln2. **B** Binding profiles of recombinant mItln2 (25 µg/mL) to microbial (left) and mammalian (right) glycan microarrays from the National Center for Functional Glycomics (NCFG). Graphical representations of top glycan hits from the glycan arrays are included. Complete microarray data are found in Supplementary Data 1. Data are shown as mean ± SD ($n = 4$ technical replicates). RFU, Relative Fluorescence Unit. **C** Glycan structures bound by mItln2 in glycan arrays. All hits contain either a terminal Gal$p$ or 6SO$_3$-Gal$p$, shown in red. **D** BLI trace of mItln2 binding to immobilized biotinylated-β-Gal$p$ at varying concentrations, with an apparent $K_d$ of 1.5 µM. **E** Dot blot analysis of immobilized MUC2, MUC5AC, and MUC5B, probed with 0.5 µM StrepII-mItln2 in the presence of Ca$^{2+}$ or EDTA. **F** Images of 5 µM StrepII-mItln2 (magenta) binding to 0.01% (w/v) fluorescently labeled MUC5AC (yellow). MUC5AC without lectin treatment served as a control. Scale bars, 20 µm. **G** Spectroscopic assay for cross-linking of 0.01% (w/v) mucins with varying amounts of mItln2, measured by the increase in absorbance at 405 nm. Schematic of the cross-linking assay is shown on the left. Data are shown as mean ± SD ($n = 3$ technical replicates). **H** Time-lapse images of 0.01% (w/v) fluorescently labeled MUC5AC (yellow) treated with 5 µM mItln2 (magenta). Binding of mItln2 was detected with Strep antibody. Scale bars, 20 µm. **I** Spectroscopic assay for cross-linking of 0.01% (w/v) lactose-functionalized glycopolymer by mItln2, measured by the increase in absorbance at 405 nm. Data are shown as mean ± SD ($n = 3$ technical replicates). Structure of the lactose-functionalized trans-poly (norbornene) is shown on the top. Results in (**D**), (**E**), and (**G**) are representative of three independent experiments. Results in (**F**), (**H**), and (**I**) are representative of two independent experiments. Source data are provided as source data file.

mucins. We first immobilized commercially available gastric mucus to nitrocellulose in a dot blot assay. Following incubation with mItln2, we observed robust binding to gastric mucus in the presence of calcium. EDTA addition abrogated binding, as expected, given that mItln2 and hItln1 are related members of the calcium-dependent X-type lectin family. These data indicate that mItln2 binds mucus by engaging mucin glycans (Supplementary Fig. 3L). To test glycan binding more directly, we added 10 mM galactose and found it could competitively displace

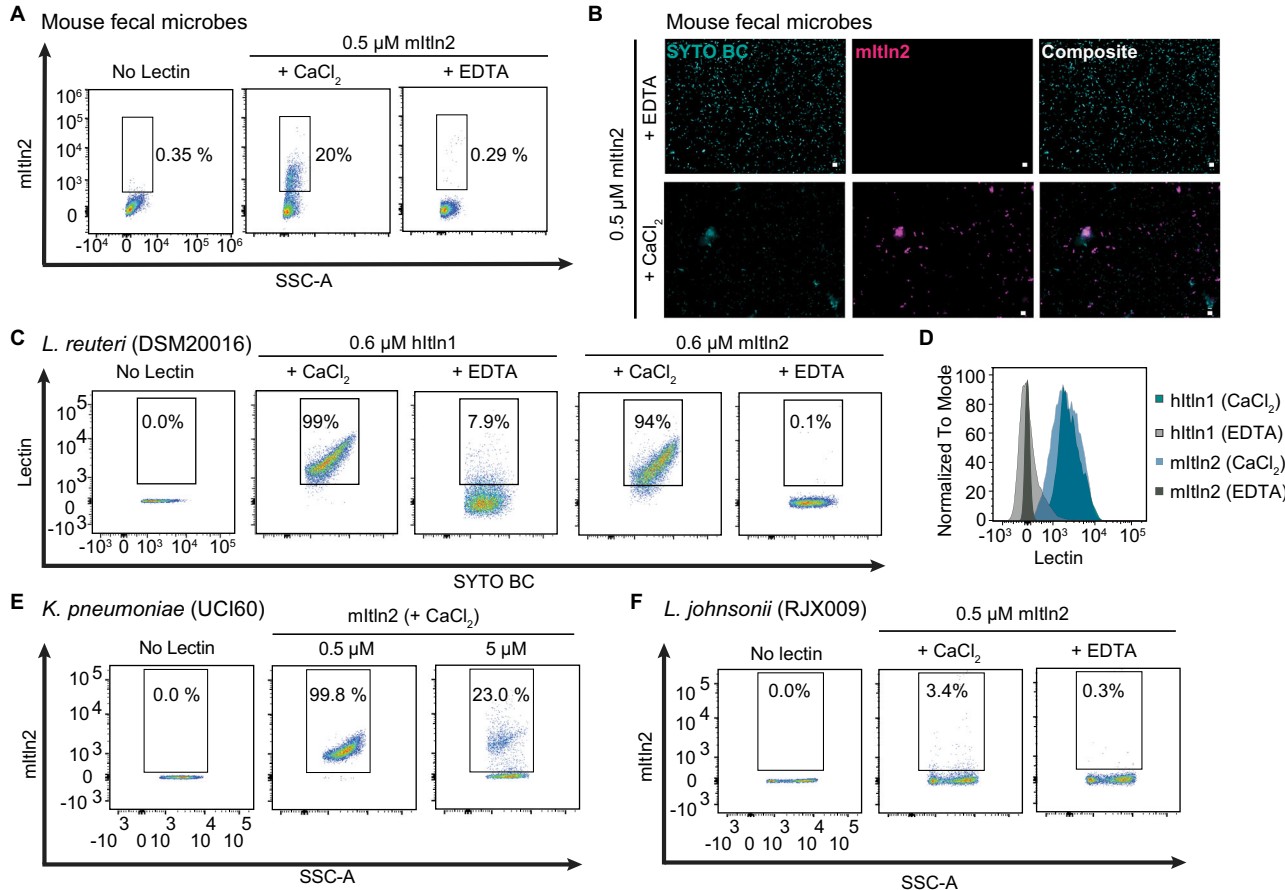

**Fig. 3 | Determination of mltln2 binding to microbes. A** Flow cytometry analysis of 0.5 μM mltln2 binding to mouse fecal samples under Ca²⁺ and EDTA conditions plotted as lectin binding (anti-Strep DY549) vs. SSC. **B** Images of murine fecal samples stained with 0.5 μM mltln2 (magenta) in the presence of EDTA (top) or Ca²⁺ (bottom) and counterstained with SYTO BC (teal). mltln2 was detected by Strep antibody. Scale bars, 10 μm. (**C, D**) Flow cytometry analysis of 0.6 μM hltln1 and mltln2 binding to *L. reuteri* under Ca²⁺ and EDTA conditions. Dot plots (**C**) show lectin binding (anti-Strep DY549) vs. nucleic acid stain (SYTO BC). Histogram (**D**) displays cell counts as a percent of the maximum signal against lectin binding in different conditions. (**E, F**) Flow cytometry analysis of mltln2 binding to *K. pneumoniae* UCI60 (**E**) and *L. johnsonii* RJX009 (**F**) under Ca²⁺ or EDTA conditions. Mltln2 was detected by Strep antibody. Unstained samples (no lectin) served as controls. Data in (**A**), (**C**), (**D**), and (**E**) are representative of three independent experiments. Data in (**B**) and (**F**) are representative of two independent experiments.

this interaction, whereas 10 mM glucose could not (Supplementary Fig. 3M). These data demonstrate that mltln2 binds mucus in a glycan-dependent manner.

We next asked if we could characterize the binding profile of mltln2 using mucins isolated from tissues that support mltln2 expression. We tested mltln2 binding to MUC2, the predominant intestinal mucin in mammals; MUC5B, a salivary and bronchial mucin; and MUC5AC, a respiratory and gastrointestinal mucin that is induced during Th2 inflammation[40]. Purified porcine MUC2, MUC5AC, and MUC5B were immobilized to nitrocellulose, and assayed for mltln2 binding in the presence of calcium ions, EDTA, or 10 mM galactose. The lectin exhibited robust binding to all three mucins. These interactions were attenuated by the addition of EDTA or galactose competitor (Fig. 2E). These data suggest that mltln2 interacts with secreted mucins at its expression sites via recognition of Gal*p* residues.

Certain lectins, such as galectins and trefoil factors, cross-link mucin to strengthen the mucus barrier[41–43]. In examining mucin-mltln2 interactions by confocal microscopy, we observed that fluorescently labeled MUC5AC co-localized in large aggregates with fluorescently labeled mltln2; this aggregation was not observed in the absence of the lectin (Fig. 2F). These observations suggest that mltln2 cross-links mucins. To examine this possibility directly, we utilized a kinetic spectroscopic assay to measure changes in light transmittance upon mixing lectin with mucin. Increasing amounts of mltln2 were titrated into solutions containing 0.01% MUC5AC, MUC5B, and MUC2 and

evaluated for changes in percent transmission over time. We observed rapid and dose-dependent clustering of mucins by mltln2, which was abrogated by the addition of galactose (Fig. 2G; Supplementary Fig. 3N). Moreover, this activity was observed at concentrations similar to those previously reported for other mucus cross-linking lectins[41]. Using time-lapse confocal microscopy, we confirmed mucin cross-linking by mltln2; lectin–mucin complexes were observed to form and grow over time (Fig. 2H; Supplementary Fig. 3O). To test whether this cross-linking depends on carbohydrate binding, we used chemically defined mucin mimetic polymers displaying pendant β-lactose (a disaccharide with terminal β-Gal*p* residue)[44]. As with mucins, mltln2 promoted robust polymer clustering (Fig. 2I), further indicating that mltln2 binds and cross-links mucins via carbohydrate recognition.

## Mltln2 is a microbicidal lectin

The epitope β-Gal*p* is found widely across taxonomic classes of bacteria[45,46]. Therefore, we hypothesized that mltln2 would bind microbes. Flow-cytometry and microscopy experiments indicate that mltln2 robustly bound mouse fecal bacteria, confirming that it interacts with microbes in native communities (Fig. 3A, B)[13]. In line with this finding, mltln2 bound numerous gram-positive and gram-negative isolates from native mouse fecal communities, including *Limosilactobacillus reuteri, Lactobacillus murinus, Lactobacillus johnsonii, Bacillus paramycoides, Mucispirillum schaedleri* and pathobionts, including *Klebsiella pneumoniae*, methicillin-resistant *Staphylococcus aureus*

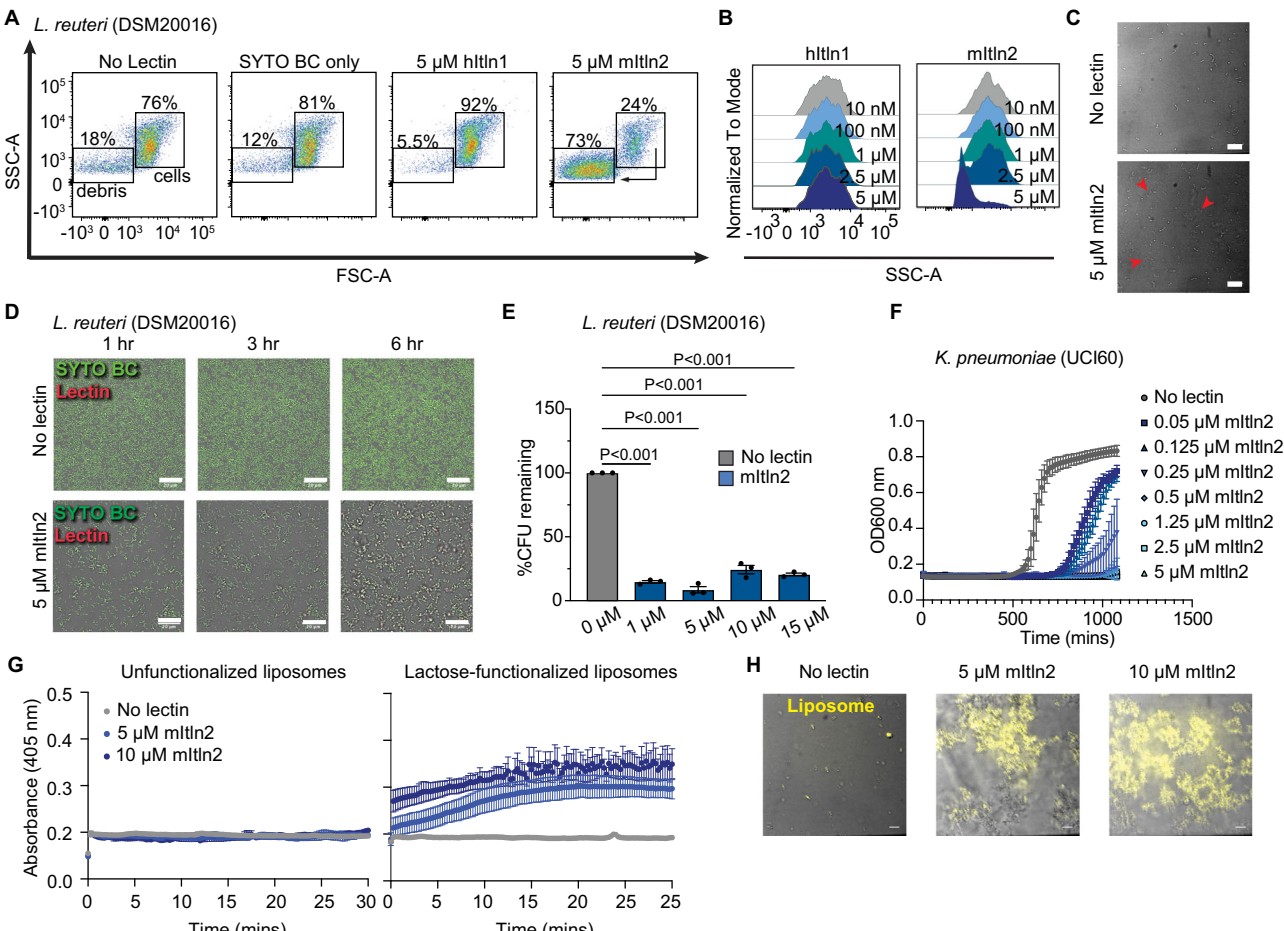

**Fig. 4 | Assessment of mltln2's impact on microbial viability. A** SSC-A vs. FSC-A analysis of *L. reuteri* DSM20016 treated with 5 μM each of hltln1 and mltln2. Untreated bacteria and SYTO BC-labeled bacteria served as controls. **B** Histogram plot displaying cell counts as a percent of the maximum signal against SSC-A for data in (**A**). **C** Brightfield microscopy of *L. reuteri* DSM20016 after 3-h treatment with 5 μM mltln2. Red arrows indicate loss of cell integrity. Untreated bacteria served as a control. Scale bars, 10 μm. **D** Time-lapse images of *L. reuteri* DSM20016 following treatment with 5 μM mltln2 (red) in the presence of Ca²⁺ and counterstained with SYTO BC (green). Scale bars, 20 μm. **E** Quantification of viable *L. reuteri* DSM20016 by dilution plating after incubation with various concentrations of mltln2 for 4 h. Data show mean ± SEM (*n* = 3 independent experiments; one-way ANOVA followed by Dunnett's multiple comparisons test). **F** Growth curve of *K.*

*pneumoniae* UCI60 in the presence or absence of mltln2 at varying concentrations. Data show mean ± SEM (*n* = 3 independent experiments). Untreated microbes served as controls. (**G**, **H**) Assessment of disruption of lactose-functionalized liposome, encapsulating Texas red-dextran dye, after treatment with mltln2 for 1 h. Spectroscopic assay (**G**) measured increase in absorbance at 405 nm following mltln2 treatment, with unfunctionalized liposomes serving as a control. Data are shown as mean ± SD (*n* = 3 technical replicates). Microscopy images (**H**) revealed dye leakage and loss of liposome integrity. Scale bars, 10 μm. Data in (**C**), (**D**), (**F**), (**G**) and (**H**) are representative of two independent experiments. Data in (**A**) and (**B**) are representative of three independent experiments. Source data are provided as source data file.

(MRSA) and *Enterococcus faecalis* (Fig. 3C–E; Supplementary Fig. 4A–E). Moreover, mltln2 binding was calcium-dependent, an indication that it is glycan-mediated. Notably, mltln2 did not bind to all tested isolates, highlighting its specificity (Fig. 3F; Supplementary Fig. 4F–J). Together, these data demonstrate that mltln2 recognizes specific microbes that colonize tissues in which mltln2 is expressed.

We next evaluated the consequences of mltln2 binding to microbes. Analysis of mltln2-bound microbes by flow cytometry revealed that mltln2 treatment caused marked shifts in forward scatter (FSC) versus side scatter (SSC) plot, in a region associated with debris (Fig. 4A, B; Supplementary Fig. 5A–E). This effect depended on the lectin concentration, with higher mltln2 concentrations resulting in increased events in the debris region for both gram-negative and gram-positive binders. This increase in debris was not observed in non-binding isolates (Supplementary Fig. 5F–K) nor when EDTA was added (Supplementary Fig. 5B, C), suggesting that the effect is glycan-dependent. Moreover, treatment with equimolar amounts of hltln1 did not show this phenotype (Fig. 4A, B). Similar alterations in FSC vs. SSC

profiles of microbes have been previously reported for other anti-microbial proteins, such as lysozyme, suggesting that mltln2 functions as an antimicrobial lectin[47]. Additionally, microscopy evaluation of mltln2-treated *L. reuteri* showed a loss of cell integrity and accumulation of ghost-like cells (empty bacterial cell envelopes) over time (Fig. 4C, D; Supplementary Movies 1 and 2). Together, these data suggest that mltln2 compromises the cellular integrity of bound bacteria and may influence cell viability.

To assess whether mltln2 has microbicidal activity, we used a colony-forming unit (CFU) assay. Bacterial isolates were grown to mid-log phase and treated with increasing concentrations of mltln2. Consistent with the flow cytometry and microscopy data, a significant reduction in the viability of *L. reuteri* and *B. paramycoides* occurred upon treatment with low micromolar concentrations of mltln2 (Fig. 4E; Supplementary Fig. 5L). These concentrations are in line with other antimicrobial proteins shown to kill microbes[48,49]. The cell-killing effect of mltln2 was confirmed using a plate reader-based growth recovery assay, which showed a significant delay in the growth recovery for

mItln2-binding *Lactobacillus* isolates following mItln2 treatment (Supplementary Fig. 6A, B). Delayed growth recovery was not observed when non-binding isolates were exposed to mItln2. The effect of mItln2 was distinct from hItln1. Specifically, isolates recognized by hItln1 had no viability loss when treated with hItln1. Thus, mItln2 possesses microbicidal activity against specific gram-positive species, an effector function lacking in the related lectin hItln1.

To test the activity of mItln2 on gram-negative pathogenic bacteria, we used *K. pneumoniae*, a common opportunistic pathogen. We observed robust mItln2 binding and a shift in the FSC vs SSC plot for *K. pneumoniae* (Fig. 3E; Supplementary Fig. 5D). Furthermore, including mItln2 in the culture media led to dose-dependent growth inhibition of *K. pneumoniae*, with a minimum inhibitory concentration of 0.3 μM (Fig. 4F; Supplementary Fig. 5M). These data suggest a mechanistic rationale for the finding that Th2 inflammation in BALB/c mice is protective against *K. pneumoniae* infection[50]. Collectively, the results indicate that mItln2 is a broad-spectrum antimicrobial protein that can target opportunistic pathogens at mucosal surfaces.

## MItln2 lyses bacterial cells

Other mammalian lectins have been implicated in microbial membrane perturbation or pore formation, processes that can kill cells and release their intracellular contents[48,49]. Given that mItln2 alters bacterial morphology and reduces viability, we postulated that mItln2 acts as a cytolytic agent by disrupting microbial membranes. In support of this hypothesis, we observed loss of DNA signal in bacteria stained with DNA intercalating dye (SYTO BC) following mItln2 treatment (Supplementary Fig. 7A; Supplementary Movie 3). We also found an increased uptake of propidium iodide, a DNA dye that cannot penetrate intact membranes (Supplementary Fig. 7B).

We employed a liposome disruption assay to better understand how mItln2 exerts its microbicidal activity (Supplementary Fig. 7C). In this approach, we generated 200 nm diameter liposomes composed of palmitoyl-oleoyl-phosphatidylcholine (POPC), 10 mol% cholesterol, and 10 mol% 1,2-dioleoyl-sn-glycero-3-phosphethanolamine-N-dibenzocyclooctyl (18:1 DBCO PE), as the latter has a functional handle to append specific glycans. Liposomes were functionalized with a β-lactose derivative (a disaccharide with a terminal β-Gal*p* residue) and treated with increasing concentrations of lectins[51]. Exposure of lactose-functionalized liposomes to mItln2 led to a concentration-dependent increase in absorbance at 405 nm, indicative of membrane disruption (Fig. 4G). This process was glycan-dependent, as adding mItln2 to liposomes lacking a glycan ligand afforded no change in absorbance. Confocal microscopy further confirmed the glycan-dependent ability of mItln2 to disintegrate the lactose-functionalized liposomes encapsulating Texas red-dextran dye (Fig. 4H). Moreover, although mItln2 bound mammalian cells, propidium iodide uptake assays demonstrate that it did not compromise cellular membrane integrity, indicating that mItln2 specifically diminishes microbial viability (Supplementary Fig. 7D, E).

Unlike other cytolytic proteins reported, the disruption of liposomes by mItln2 depended on glycan binding and occurred at physiological pH (7.4) and osmolarity (150 mM NaCl). The liposome disruption and cell-killing activity did not require an acidic pH or a low salt concentration (<50 mM). These findings reveal that the powerful cell-killing ability of mItln2 synergizes with its ability to bind and cluster mucins. Thus, mItln2 has dual roles during Th2 inflammation: it functions by cross-linking host mucins and by binding and killing microbes.

## Human intelectin-2 (hItln2) is a galactose-binding antimicrobial lectin

Human intelectin-2 (hItln2) is constitutively expressed by Paneth cells in the small intestine and is detectable in the lungs[19,52]. A recent study demonstrated that hItln2 expression is even higher than hItln1 in

healthy small intestinal tissue and that aberrant expression of hItln2 is observed in both the ileum and colon of CD patients[19]. While Th2 stimuli have been shown to upregulate hItln1 expression in the lungs, Th2 inflammation-dependent expression of hItln2 has not been directly demonstrated[24]. Our analysis of the promoter region (−2000 base pairs to −1 base pair upstream of transcription start site) of hItln2 indicated potential STAT6-binding motif; however, further experimental validation is required to confirm Th2-dependent upregulation of hItln2 (Supplementary Fig. 8A). Nevertheless, consistent with its constitutive intestinal secretion, hItln2 is readily detected in human fecal samples (Supplementary Fig. 8B). However, as with mItln2, the ligand specificity and biological function of hItln2 have not been clear.

To characterize hItln2, we expressed and purified recombinant StrepII-tagged hItln2. The production of recombinant hItln2 required optimization. We made two changes: an introduction of a glycosylation site (A177S) that is conserved in hItln1 and mItln2 and the removal of an unpaired cysteine near the C-terminus (C311G) that does not participate in trimerization and binding data suggests is amenable to substitution (Supplementary Fig. 8C)[53]. These modifications improved solubility and reduced aggregation, yielding soluble, stably folded protein with a melting temperature of approximately 54 °C (Supplementary Fig. 8D). HItln2 formed covalent trimers and higher-order oligomers, as observed for native hItln2 in small intestinal tissue (Supplementary Fig. 8E, F)[19]. Consistent with our experimental data, the AlphaFold 3-predicted structure of hItln2 is a trimer with a similar ligand binding domains as hItln1 and conserved intermolecular disulfide linkages between C42 and C60 residues of the neighboring monomers (Supplementary Fig. 8G).

Like mItln2, the ligand binding site of hItln2 lacks the aromatic box present in Itln1 (Fig. 5A; Supplementary Fig. 1B). In hItln2, a serine at position 309 replaces the tyrosine at the corresponding site in hItln1. We reasoned that this more open binding pocket might accommodate pyranose sugars, as observed with mItln2. We next investigated if hItln2 and mItln2 share glycan targets. Using BLI to evaluate the binding of hItln2 with immobilized mono- and di-saccharides, we observed that hItln2 indeed binds mItln2 ligands (β-Gal*p* and β-6SO$_3$ LacNAc) but not the Itln1 ligand (β-Gal*f*) (Fig. 5B–D). These data are supported by a recent report indicating a monomeric hItln2 variant lacking disulfides binds to Gal*p*[53]. The apparent dissociation constant of hItln2 for β-Gal*p* was approximately 3.9 μM (Supplementary Fig. 8H, I). Thus, mItln2 and hItln2 share common ligands. Moreover, the microbial glycan array analysis for hItln2 yielded hits containing Gal*p*, but also primary amine-containing substituents (e.g., phosphoethanolamine (PEt)), suggesting that terminal amine-containing substituents may serve as an additional microbial ligand for hItln2 (Fig. 5E, F; Supplementary Data 2). These overlapping yet distinct ligand recognition properties could be due to other differences in the binding site residues of mItln2 and hItln2 (mItln2/hItln2: E274/Q286, A288/W300, and Y297/S309).

Because mItln2 and hItln2 bind to β-Gal*p*, we reasoned that they might interact with similar physiological substrates. Supporting this hypothesis, we found that hItln2 binds intestinal mucins, MUC2, and MUC5AC in a calcium ion-dependent manner (Fig. 5G, H). Spectroscopic analysis of hItln2−mucin interactions showed mucin cross-linking by hItln2, but hItln2 activity was more challenging to detect via microscopy (Supplementary Fig. 8J–L). The minor differences in mucin selectivity observed between mItln2 and hItln2 may be due to differences in mucin glycosylation patterns (density, modification, and accessibility) or the influence of adjacent glycans on lectin selectivity. For example, porcine MUC2 displays glycans reported to be shorter and less extended than in MUC5AC and MUC5B[54]. Moreover, mItln2's non-covalent oligomerization state (monomer, dimer, trimer, hexamer) as well as its pronounced binding to extended, branched LacNAc array ligands compared to disulfide-bonded hItln2 may also contribute to the observed differences.

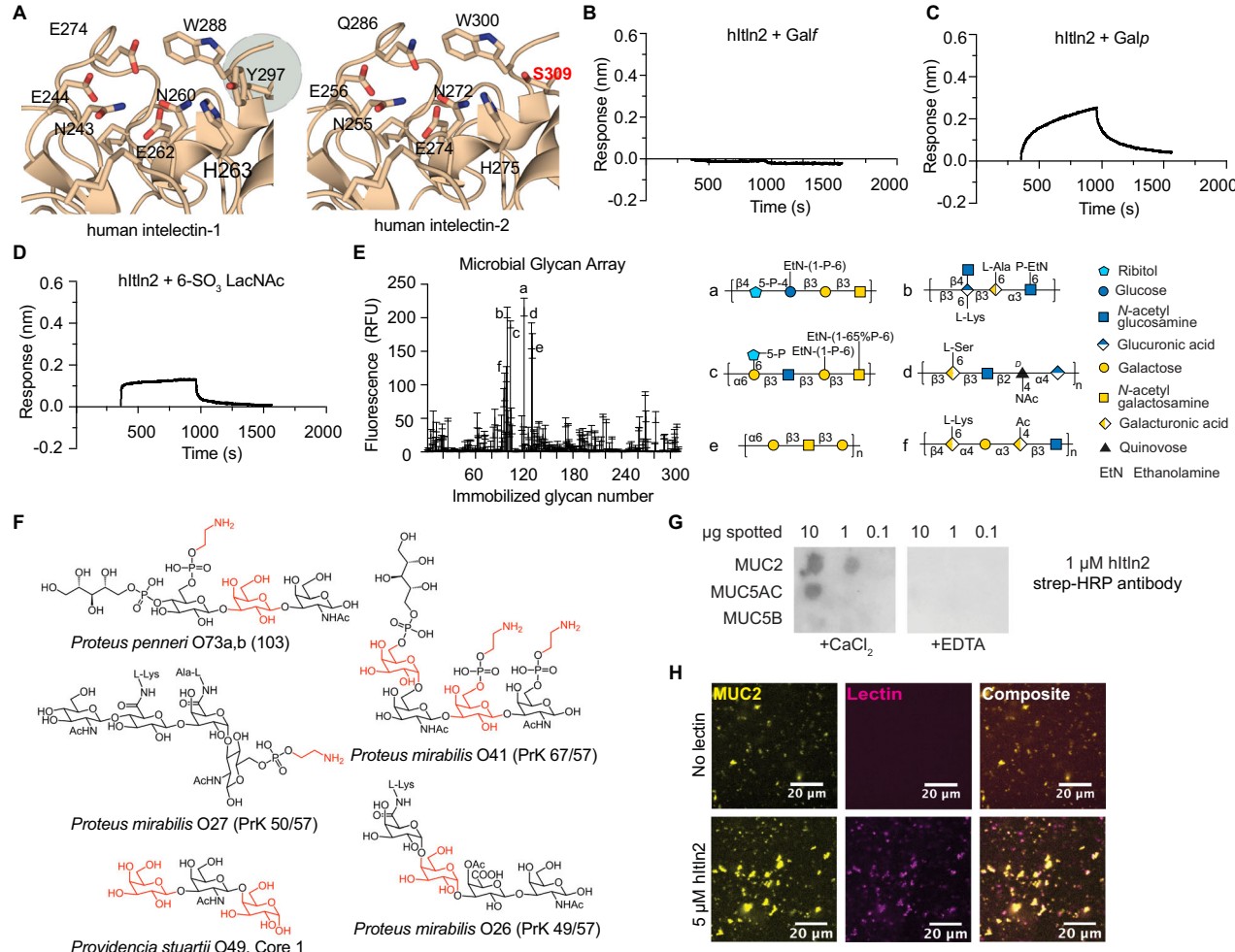

**Fig. 5 | Characterization of glycan specificity for hItln2. A** Comparison of carbohydrate recognition domain of hItln1 (top, PDB ID 4WMY) and hItln2 (bottom, predicted model), showing conserved calcium coordination residues and a serine replacing tyrosine at position 309 in hItln2. (**B**–**D**) BLI traces of hItln2 binding to immobilized biotinylated-β-Gal*f* (a hItln1 ligand) (**B**), biotinylated-β-Gal*p* (**C**), and biotinylated-6SO₃ LacNAc (**D**). In (**B**) and (**C**), 1.5 μM protein was used. In (**D**), 10 μM protein was used. Data were normalized by background subtraction, using biotin-loaded streptavidin. **E** Binding of recombinant hItln2 (25 μg/mL) to a microbial glycan microarray from NCFG. Graphical representations of top glycan hits from the glycan arrays are included. Complete microarray data are found in

Supplemental Data 2. Data are shown as mean ± SD (*n* = 4 technical replicates). **F** Glycan structures bound by hItln2 in glycan arrays. All hits contain either a Gal*p* or ethanolamine, shown in red. **G** Dot blot analysis of MUC2, MUC5AC, and MUC5B, spotted on the membrane and probed with 1 μM hItln2 in the presence of Ca²⁺ or EDTA. Binding of hItln2 was detected using a Strep-HRP antibody. **H** Images of 5 μM hItln2 (magenta) binding to 0.01% (w/v) fluorescently labeled MUC2 (yellow). Binding of hItln2 was detected with Strep antibody. Scale bars, 20 μm. Results in (**B**), (**C**), (**D**) and (**G**) are representative of three independent experiments. Result in (**H**) is representative of two independent experiments. Source data are provided as source data file.

Like mItln2, hItln2 also recognized microbes within a human fecal sample (Fig. 6A, B) and a range of gram-positive and gram-negative bacterial isolates that includes the opportunistic pathogens *Escherichia coli* and *Enterococcus faecalis*. These cell-binding properties were inhibited by the chelator EDTA (Supplementary Fig. 9A–F), indicating that hItln2 interacts with glycans through its calcium ion. Unlike mItln2, hItln2-treated microbes formed large cellular aggregates, which could be disrupted with EDTA, findings consistent with glycan-dependent agglutination (Supplementary Fig. 9B, D).

We investigated if hItln2 functions as an antimicrobial protein based on several key characteristics: (1) abundant expression at sites of microbial colonization; (2) secretion from canonically anti-microbial Paneth cells; and (3) cationic nature (isoelectric point of 8.3), and (4) lengthened N-terminal sequence (relative to hItln1) that is predicted to form an alpha-helix (Supplementary Fig. 9G). We first evaluated if hItln2 could bind the pathobiont *Staphylococcus aureus*, a common gram-positive opportunistic pathogen. Increasing evidence suggests that *S. aureus* can colonize the human gut and, under the right conditions, can disseminate to other host tissues via translocation across

the intestinal mucosa and epithelium[55]. We reasoned that hItln2 might encounter this organism in the gastrointestinal tract. Consistent with this hypothesis, hItln2 showed robust binding to and pronounced agglutination of methicillin-resistant S. aureus (MRSA) by flow cytometry and microscopy (Fig. 5C, D, Supplementary Fig. 9H).

To evaluate the antimicrobial properties of hItln2, we performed CFU assays on hItln2-treated *S. aureus*. To this end, MRSA was grown to mid-log phase and then exposed to increasing amounts of hItln2. Cells were then vigorously mixed, serially diluted in assay buffer with thorough mixing between dilutions, plated on growth substrate, and left overnight at 37 °C before counting colonies. As we found for mItln2, hItln2 treatment decreased CFU counts for MRSA, indicating reduced bacterial viability (Fig. 6E). This result was also obtained after treatment of other gram-positive bacteria recognized by hItln2 (Supplementary Fig. 9I). HItln2 also bound and inhibited the growth of the gram-negative pathobiont *K. pneumoniae* in a dose-dependent manner, with minimum inhibitory concentration of 0.3 μM (Fig. 6F, G, Supplementary Fig. 9J). Together, these data suggest that hItln2 inhibits microbial viability and growth through the antimicrobial effector

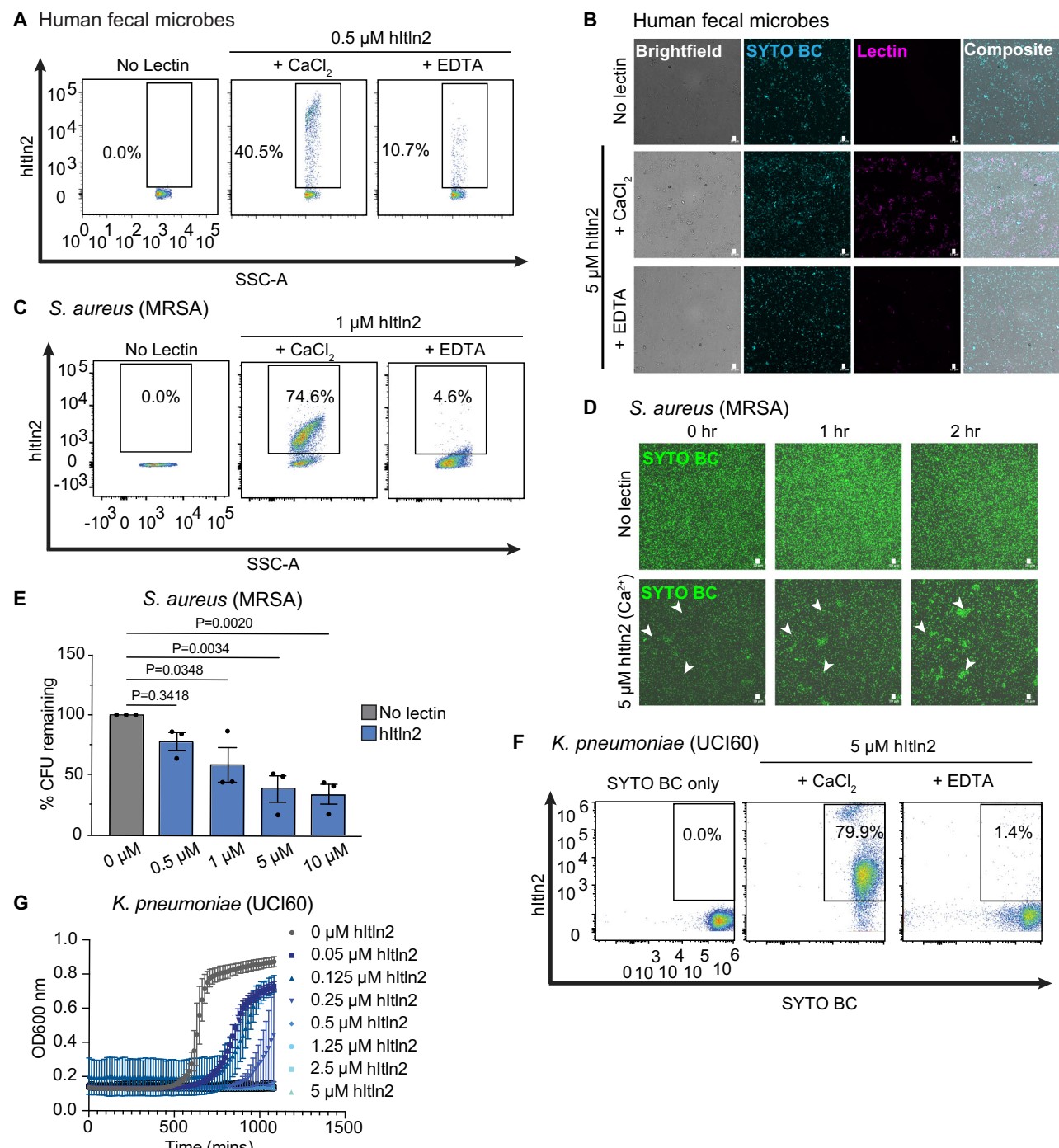

**Fig. 6 | Evaluation of hItln2 binding to microbes and its impact on microbial viability. A** Flow cytometry analysis of 0.5 μM hItln2 binding to human fecal samples under $Ca^{2+}$ and EDTA conditions, plotted as lectin binding (Streptactin DY549) vs. SSC. **B** Microscopy images of human fecal samples stained with 5 μM hItln2 (magenta) in the presence of $Ca^{2+}$ or EDTA and counterstained with SYTO BC (teal). hItln2 was detected by Streptactin. Scale bars, 10 μm. **C** Flow cytometry analysis of 1 μM hItln2 binding to *S. aureus* MRSA under $Ca^{2+}$ and EDTA conditions, plotted as lectin binding (anti-Strep DY549) vs. SSC. Treatment with SYTO BC and Strep antibody (without lectin) served as a control. **D** Time-lapse images of *S. aureus* MRSA treated with 5 μM hItln2 in the presence of $Ca^{2+}$ and counterstained with SYTO BC (green). Examples of cell agglutination are indicated with white

arrowheads. Untreated bacteria served as control. Scale bars, 10 μm. **E** Viable *S. aureus* MRSA quantified by dilution plating after incubation with various concentrations of hItln2 for 4 h. Data represent mean ± SEM (n = 3 independent experiment; one-way ANOVA followed by Dunnett's multiple comparisons test). **F** Flow cytometry analysis of hItln2 binding to *K. pneumoniae* UCI60 under $Ca^{2+}$ or EDTA conditions, detected by Streptactin. Treatment with SYTO BC (without lectin) served as a control. **G** Growth curve of *K. pneumoniae* UCI60 in the presence or absence of hItln2 at varying concentrations. Data show mean ± SEM (n = 3 independent experiments). Data in (**A**), (**B**), (**C**), (**D**), and (**F**) are representative of three independent experiments. Source data are provided as source data file.

function of agglutination. Like mItln2, hItln2 did not compromise the membrane integrity of mammalian cells (Supplementary Fig. 10A, B).

Expression of hItln2 in healthy subjects is primarily restricted to the small intestine[19]. Our initial characterization of hItln2 activity was performed under ileum-like conditions: neutral pH and physiological ionic strength. However, hItln2 is present in the colons of patients with colonic CD and UC due to Paneth cell metaplasia[19]. The colonic environment in CD and UC patients can range from acidic to neutral pH (pH 4–7) and has altered ionic strength owing to aberrations in electrolyte uptake and secretion during inflammation[56–58]. Moreover, other small intestinal-derived antimicrobial proteins are reported to be active only under acidic pH and low ionic strength[48,59–61]. To address these variables, we investigated hItln2 activity at low pH and ionic strength.

Even under these stress conditions, hItln2 remained well-folded and retained binding to β-Gal$p$ and β-6SO$_3$ LacNAc (Fig. 7A; Supplementary Fig. 11A, B). We also confirmed that hItln2 bound to mucins and microbes under these conditions (Fig. 7B–D; Supplementary Fig. 11C–E); however, EDTA addition did not abrogate these interactions, suggesting a shift to an alternate binding mode. Moreover, under these conditions, hItln2 mediated rapid and robust formation of mucin and microbial aggregates (Fig. 7E, F; Supplementary Fig. 11F, G). Similarly, CFU analysis showed a dramatic reduction in the growth of the pathobionts, *S. aureus*, *B. paramycoides*, and *E. coli*, confirming that the agglutination-mediated antimicrobial function of hItln2 was enhanced under conditions that represent intestinal inflammation (Fig. 7G, H; Supplementary Fig. 11H–J).

## Discussion
Both mItln2 and hItln2 have been linked to infection and disease[6,7,19,33]; however, their specific effector functions were unclear. Here, we identified these proteins as galactose-binding lectins with protective functions in host defense against microbes. Specifically, our findings reveal that mItln2 and hItln2 recognize Gal$p$-containing glycans to confer two key functional properties: mucin cross-linking and antimicrobial activity (Supplementary Fig. 12).

The sequence similarity between Itln1 and Itln2 indicates that they share similar overall structures. Itln1 binds exocyclic vicinal diols on microbial carbohydrates, like Gal$f$, through a calcium ion, with aromatic residues surrounding the saccharide. However, Itln1 does not bind the Itln2 ligand Gal$p$, which lacks an exocyclic vicinal diol. A critical difference between the Itln1 and predicted Itln2 binding sites is the absence of an aromatic residue in Itln2. Substitution at this position removes the aromatic box and results in a more open binding site, which could facilitate binding of Itln2 to vicinal diols within the pyranose ring of galactose. These findings highlight the versatility of the intelectin fold, where changing a single residue in the binding site alters glycan specificity. This feature suggests intelectins are excellent platforms for engineering lectins with unique binding specificities.

In mice, Th2 cytokines elicit the conditional expression of mItln2, suggesting its inducibility in response to pathogens capable of breaching mucus barriers. The ability of mItln2 to cross-link mucins and kill microbes highlights its protective role in strengthening the mucosal barrier and limiting the spread of infection during Th2 immunity. Notably, Th2 cytokines are also known to drive goblet cell hyperplasia and enhance mucus secretion, suggesting a co-defense mechanism for handling the Th2 inflammatory conditions[62]. The ability of mItln2 to kill both pathogens and commensals may explain its limited and inducible expression in healthy tissues, where commensals are essential for maintaining host health. In contrast to some antimicrobial lectins that only exert antimicrobial activity under low pH and low salt conditions and in a glycan-independent manner, the bactericidal effect of mItln2 is distinctive due to its glycan dependence and efficacy under physiological pH and osmolarity conditions[48]. These characteristics underscore the selectivity and robust

antimicrobial function of mItln2, which is necessary for its activity at sites beyond the gastrointestinal tract, such as the lungs.

Although it has been argued that hItln2 is not an ortholog of mItln2[19], these homologous proteins share the ability to cross-link host mucins and bind microbes at pH and osmolarity of their native expression environment, which is contrast to other antimicrobial proteins secreted in the same tissues. Moreover, we found that the antimicrobial function of hItln2 involves microbial agglutination and inhibition of bacterial growth. Microbial agglutination was not observed with mItln2 but is a functional activity of other antimicrobial proteins, including zymogen granule proteins and human α-defensin-6[63–65]. Under homeostatic conditions in the small intestine, constitutively expressed hItln2 might serve to prevent the overgrowth of microbes via agglutination. However, in environments with higher acidity and altered ionic strength, as observed in CD and UC patients, ligand recognition by hItln2 occurs independent of calcium, and its mucin-cross-linking and antimicrobial activities are significantly enhanced. This change in activity suggests that hItln2 may act as a sensory switch. Moreover, these findings highlight how the lectin function can be altered by inflammation. Still, it remains unclear whether the enhanced activities of hItln2 under pathological conditions help limit inflammation and reestablish mucosal health or alter the microbial composition and exacerbate inflammation. Further investigations are needed to distinguish between these roles.

While mItln2 and hItln2 both bound mammalian cells, they did not affect their viability, which suggests a possible role of differences in membrane composition or the ligand presentation between mammalian and microbial cells on lectin activity. Such behavior has also been reported in other antimicrobial proteins, such as Galectin-7, that inhibit microbial growth but do not kill mammalian cells despite binding[49]. Additionally, we reason that several other factors could contribute to host versus microbial discrimination by Itln2 in vivo. Firstly, Itln2 is secreted in environments rich with microbes and mucins, and these components may compete for binding or create local concentration gradients that favor targeting of microbes and mucins over mammalian cell surfaces. Such competition would limit Itln2's access to host epithelial cell surfaces. Additionally, microbial surface glycans and mucins may present β-D-galactopyranose residues at a higher density, thereby facilitating preferential recognition of intelectin-2 to microbes and mucins.

We posit that intelectins, which are highly conserved across chordates, represent an evolutionarily ancient solution to host defense that predates the adaptive immune system. In contrast to canonical antimicrobial proteins that directly disrupt bacterial membranes through electrostatic interactions, intelectins function through more specific targeting of distinct carbohydrate epitopes. Additionally, the family can participate in a range of effector functions, including immune cell recruitment and interacting with mucus, properties that distinguish them from classic antimicrobial proteins[1].

The dual functions of Itln2 suggest it evolved to address a specific challenge in barrier immunity: the generation of antimicrobial barriers that are both structurally robust and functionally active. This need is even more critical when tissues are undergoing active remodeling and repair during type 2 inflammation. Our data support a model in which Itln2 protects the host against microbial threats: a defensive role in which it binds mucins to reinforce the mucosal environment and an offensive role in which it reduces bacterial burden via its antimicrobial activities. These findings advance our understanding of intelectins as crucial players in host–microbe interactions, with implications for maintaining mucosal health in both homeostatic and diseased states.

## Methods
The research in this paper complies with all ethical regulations set by the Institutional Review Board (MIT) and the Institutional Animal Care and Use Committee (IACUC, UC-Davis). Human stool samples used in

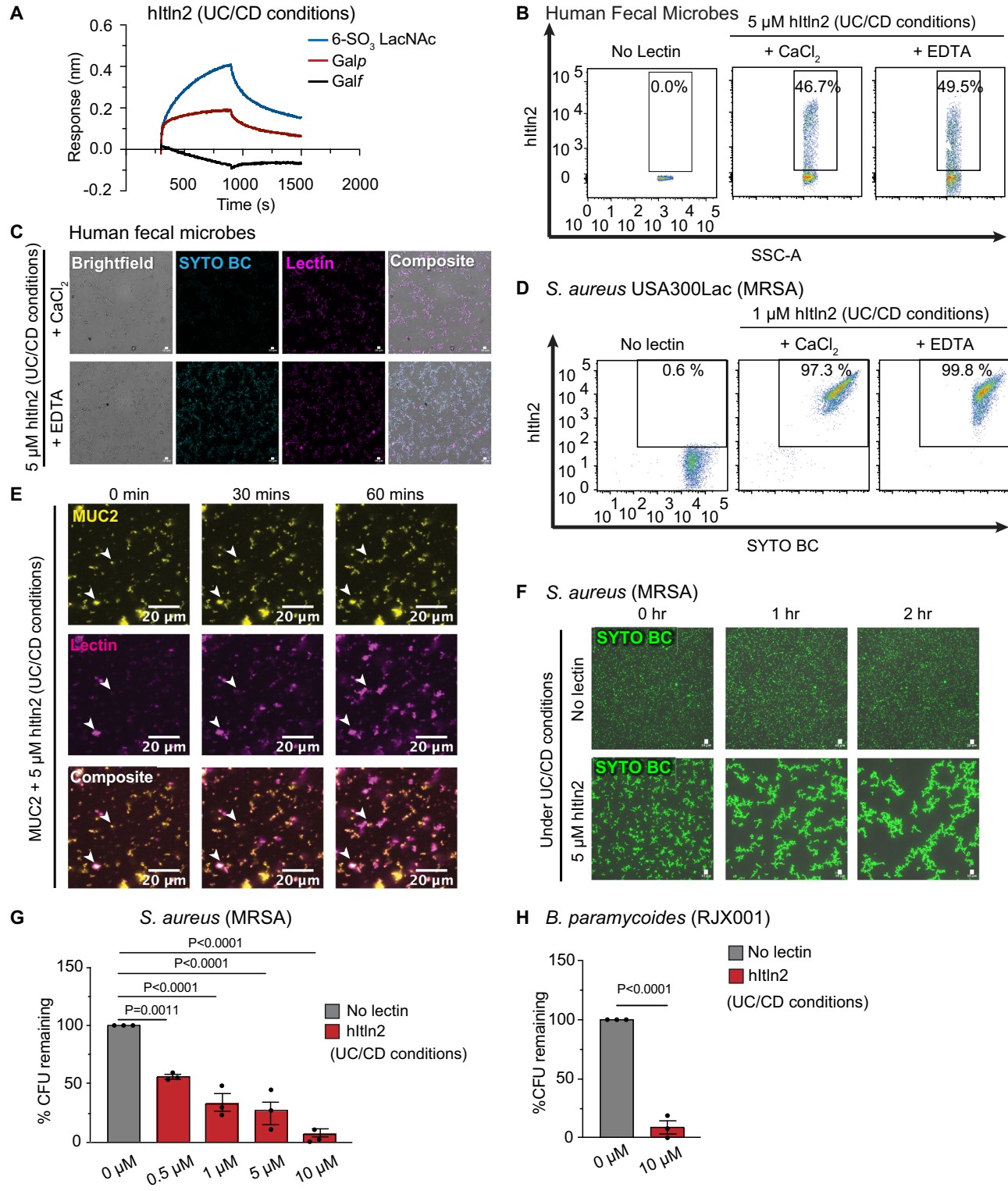

**G** *S. aureus* (MRSA)

**H** *B. paramycoides* (RJX001)

the flow cytometry test of human intelectin-2 were obtained under a protocol approved by the Massachusetts Institute of Technology [Institutional Review Board (IRB) protocol ID no. 1510271631]. The participants provided informed consent, and all experiments complied with the review board regulations. Human stool samples were collected for a previously published study[13]. Banked stool sample from the aforementioned study was used in our study. Patient samples were selected based on diagnosis regardless of sex and gender. All animal experiments were approved by the Institutional Animal Care and Use Committee at the University of California, Davis.

**Congenic C57BL/6NTac.BALB/cAnNTac-Itln1:6 (B6.C-Itln1-6) mouse model**

Male BALB/cAnNTac mice, encoding a full *Itln1-6* locus[18], were crossed with female C57BL/6NTac mice that encode a single intelectin gene, *Itln1* (parent strains obtained from Taconic Biosciences, Germantown, NY). Male offspring, positive for the *Itln1-6* locus (F1/N1: heterozygous), were crossed with pure C57BL/6NTac female mice for an additional nine generations (N10: ~99.9% C57BL/6NTac background). Experimental tissues and enteroids were derived from 10-week-old

**Fig. 7 | Evaluation of hItln2 binding to mucin and microbes at low pH and low salt (UC/CD conditions). A** BLI trace of 1.5 µM hItln2 binding to immobilized biotinylated-β-Gal$f$, biotinylated-β-Gal$p$, and biotinylated-6SO$_3$ LacNAc under low pH and salt conditions (UC/CD conditions). Data were normalized by background subtraction, using biotin-loaded streptavidin. (**B**, **C**) Flow cytometry (**B**) and microscopy (**C**) of 5 µM hItln2 binding to human fecal samples in Ca$^{2+}$ and EDTA conditions under low pH and salt concentrations. Dot plot (**B**) displays lectin binding (Streptactin DY549) vs. SSC. hItln2 was detected by Streptactin. **D** Flow cytometry of 1 µM hItln2 binding (detected by Streptactin) to *S. aureus* MRSA in Ca$^{2+}$ and EDTA conditions under low pH and salt concentrations. Treatment with SYTO BC and Streptactin (without lectin) served as a control. **E** Time-lapse images of 0.01% (w/v) fluorescently labeled MUC2 (yellow) treated with 5 µM hItln2 (magenta) under low pH and low salt conditions. Binding of hItln2 was detected with Strep antibody. Examples of mucin cross-linking are indicated with white arrowheads. Scale bars, 20 µm. **F** Time-lapse images of *S. aureus* MRSA treated with 5 µM hItln2 in the presence of Ca$^{2+}$ and counterstained with SYTO BC (green) under low pH and low salt conditions. Untreated bacteria served as a control. Scale bars, 10 µm. **G** Viable *S. aureus* MRSA quantified by dilution plating after incubation with various concentrations of hItln2 for 4 h under low pH and low salt conditions. Data show mean ± SEM ($n = 3$ independent experiments; one-way ANOVA followed by Dunnett's multiple comparisons test). **H** Quantification of viable *B. paramycoides* RJX001 by dilution plating after treatment with 10 µM hItln2 for 4 h at low pH and low salt conditions. Data represent men ± SEM ($n = 3$ independent experiments; unpaired two-tailed t-test). Data in (**A**) and (**E**) are representative of two independent experiments. Data in (**B**), (**C**), (**D**), and (**F**) are representative of three independent experiments. For complete data, see the source data file.

heterozygous mice. Animals were humanely euthanized under deep anesthesia with ketamine/xylazine (100/10 mg/kg).

### Genotyping
DNA was isolated from tissue samples by HotSHOT method[66]. Briefly, samples were incubated in alkaline lysis buffer (25 mM NaOH, 0.2 mM EDTA, pH 12.0) overnight in a water bath at 65 °C. Following digestion, an equivalent volume of neutralization buffer (40 mM Tris-HCl, pH 5.0) was added to the digest solution. The *Itln1-6* haplotype was identified using primers designed to amplify both *Itln1* and *Itln6*, where the *Itln1* allele (i.e., wild-type C57BL/6N; deletion) generates a 305 bp product, and the *Itln6*-containing allele generates an 88 bp product. Genotyping primers: F: 5′-TATTCCTGTCTCAGCTCCTAG-3′, and R: 5′-GTCA-CAGGTAAAKCCAGAAGG-3′ (K = G or T). Polymerase chain reaction (PCR) was performed using an Applied Biosystems thermocycler (Waltham, MA). The PCR products were amplified using *Taq* DNA polymerase (New England Biolabs, Cat# M0273S). Thermocycler conditions: one cycle: 95 °C (30 s), 35 cycles: (95 °C [30 s], 62 °C [30 s], and 70 °C [30 s]), and additional extension for 5 min at 70 °C.

### Enteroid culture
Small intestinal crypts were isolated as previously described with minor modification[67]. Briefly, 10 cm of distal small intestine was flushed with Dulbecco's PBS (DPBS), dissected longitudinally, scraped with a glass coverslip to remove villi, and minced into ~1 cm pieces. Intestinal fragments were incubated at 4 °C under gentle rocking (60 rpm) for 45 min in ice-chilled 2 mM EDTA (DPBS) dissociation solution. The intestinal pieces were allowed to settle at the bottom of the conical tube by gravity, and the dissociation solution was decanted. Following a second 45 min incubation, intestinal pieces were vigorously shaken for ~3 min in dissociation solution to release intestinal crypts. The resulting crypt-containing suspension was passed through a 70 µm filter to exclude villi fragments, pelleted, and enumerated. The crypts were plated on 24-well Costar® culture dishes (StemCell Technologies, Cat# 38017) as 50 µL suspensions of 50% v/v growth factor reduced matrigel (Corning, Cat# 356231) in murine Intesticult™ Organoid Growth Media (StemCell Technologies, Cat# 06005), supplemented with 50 µg/mL gentamycin (Sigma-Aldrich, Cat# G1397). Enteroids were passaged every 12 days following standard procedures (StemCell Technologies, document# 28223), and complete media was replenished every three days. For cytokine treatment, enteroids were cultured for nine days prior to 72 h treatment with recombinant (PeproTech, Cranbury, NJ) murine IL-4 (Cat# 214-14, 20 ng/mL), IL-13 (Cat# 210-13, 10 ng/mL), or IL-22 (Cat# 210-22; 20 ng/mL).

### RNA extraction, cDNA synthesis, and RT-qPCR
RNA extraction from intestinal tissue was performed using the guanidine thiocyanate/cesium chloride gradient method, as previously reported[18,68]. For enteroids, media was removed from the 24-well culture plates following cytokine treatment, and 500 µL of TRIzol™ reagent (ThermoFisher Scientific, Cat#15596026) was added directly to the Matrigel domes. RNA was extracted as outlined by the manufacturer with minor modifications. Following RNA precipitation and decanting of isopropyl alcohol, resulting pellets were resuspended in 250 mL of 70% molecular grade ETOH (30% v/v DEPC H$_2$O) with 10 mL of 3 M sodium acetate, and incubated overnight at −80 °C. Thereafter, the RNA was again pelleted by centrifugation, washed with 80% ETOH, and resuspended in DEPC H$_2$O to determine concentration by ultraviolet absorbance (Nanodrop™) spectroscopy. cDNA synthesis was performed as previously reported, using 1–3 mg of isolated RNA with the SuperScript™ III First-Strand Synthesis kit (ThermoFisher Scientific, Cat# 18080051), followed by column purification using the QIAquick PCR purification kit (Qiagen, Cat# 28104)[18]. RT-qPCR was performed using a Roche Lightcycler® 2.0 under conditions parallel to those previously reported[18,68]. Primers: *Actb* forward 5′-GGCTGTATT CCCCTCCATCG-3′, *Actb* reverse 5′-CCAGTTGGTAACAATGCCATGT-3′. *Itln1* forward 5′-ACCGCACCTTCACTGGCTTC-3′, *Itln1* reverse 5′-CCAAC ACTTTCCTTCTCCGTATTTC-3′. *Itln*-common forward 5′-GCCTCAG-CAGAGAAAGGTTCC-3′, *Itln*-common reverse 5′-GAAGGTCTGGTA-GATGACACCATTC-3′. *Clca1* forward 5′-CCTGACTCCTGACTTCTTAGC-3′, *Clca1* reverse 5′-TGAACACCTCACTGCTTGG-3′. *Reg3g* forward 5′-CCTCAGGACATCTTGTGTC-3′, *Reg3g* reverse 5′-TCCACCTCTGTTGG GTTCA-3′[18,68].

### Next-generation sequencing of Itln paralogs
A subset of PCR reactions, generated by RT-qPCR using the *Itln*-common primers, were pooled (B6.C-*Itln1-6*: distal small intestine; $n = 3$ animals, and enteroids; $n = 6$ independent samples/treatment), column purified (QIAquick PCR purification kit) and submitted for Illumina sequencing (GENEWIZ Amplicon-EZ, Azenta Life Sciences). Sequence data (i.e., unique reads) were binned and then aligned to the reference mRNA for the individual *Itlns* (*Itln1*, *Itln2*, *Itln3*, *Itln4*, *Itln5*, and *Itln6*) retrieved from the BAC contig of the 129S7 strain (Gene Bank #HM370554), which like BALB/c (parental donor strain for the B6.C-*Itln1-6* congenic mouse model), encodes a full *Itln1-6* locus[18]. Percentage total and individual *Itln1-6* read count (Fig. 1C, D, Supplementary Table 1) was generated following two selective filters of the raw data: (1) 283 nt (i.e., expected PCR product size), and (2) the minimum recorded unique read count was set at 2000, representing ≤0.2% of total reads (median total read count of experimental groups = 1.23 million; Supplementary Table 1).

### Cloning of StrepII-tagged mouse intelectin-2 and StrepII-tagged human intelectin-2
Forward primer (5′GCGTTTAAACTTAAGCTTCACCATGACC-CAACTGGGCTTCCTG3′) and reverse primer (5′CCACCA-CACTGGACTAGTGGATCCTCATTAGCGATAAAACAGAAGCACAGC3′) were used to amplify strep-mItln2 (Accession number AAO60215, https://www.uniprot.org/uniprotkb/Q80ZA0/entry) from pFastBac-Strep-mItln2 vector (Kiessling lab, unpublished). Forward primer (5′-AGTTAAGCTTCACCATGCTGTCCATGCTGAGGACAATGACC-3′) and

Reverse primer (GCTCGGATCCTCATTATCTATAGAACAAGAGTACA GCCGCCTCCGTT) were used to amplify hItln2 cDNA (AY358905) (Kiessling Group, unpublished). The resulting PCR products were inserted into pcDNA4 using Gibson ligation. For hItln2, StrepII-tag was inserted using two-step Quikchange mutagenesis with the following primers: (5'GCAGCAGCCTCTTCTggagccatccgcagtttgaaaagtcttctCTT-GAGATGCTCTCG3') and (5'CGAGAGCATCTCAAGagaagactttttcaaa ctgcggatggctccaAGAAGAGGCTGCTGC3'). Plasmid sequences were verified using Sanger sequencing (Quintara Biosciences) with pCMV forward (5'CGCAAATGGGCGGTAGGCGTG3') and BGH reverse primers (5' CTAGAAGGCACAGTCGAGG 3').

## Recombinant protein expression and purification

Recombinant hItln1, mItln2, and hItln2 with N-terminal Strep-tag II were expressed by transient transfection of suspension-adapted HEK293 cells, following established procedures[12]. HEK293 cells were maintained at a density of $1 \times 10^6$ cells/mL in DMEM medium (ThermoFisher, Cat# 11995) supplemented with 10% heat-inactivated FBS, 1x penicillin-streptomycin, 1x L-glutamine, and 1x non-essential amino acids in spinner flasks at 37 °C, 5% $CO_2$. Cells were passaged every 2–3 days, with the addition of Pluronic-F68 (ThermoFisher). Transfection was carried out at $1 \times 10^6$ cells/mL in the growth medium using Lipofectamine 2000 (ThermoFisher, Cat# 11668030) as per the manufacturer's protocol. Six hours post-transfection, the culture medium was replaced with FreeStyle F17 expression medium (ThermoFisher, Cat# A1383501), supplemented with 1x penicillin-streptomycin, 1x L-glutamine, 1x non-essential amino acids. Transfected cells were cultured for up to 3 days, and the expression medium was harvested by centrifugation and sterile filtration.

To verify protein expression, expression media was combined with 6x Laemmli buffer containing dithiothreitol, boiled at 95 °C for 10 min, and run on 4–15% TGX gel (BioRad). Proteins were transferred to PVDF membrane, blocked in 5% milk in TBST (Tris-Buffered Saline with Tween-20), and blotted for the presence of mItln2 using 1:5000 rabbit anti-Itln2 (Proteintech, Cat# 11770-1-AP) and goat-anti-rabbit HRP (1:10,000, Jackson, Cat# AB_2313567). hItln2 was detected on protein expression blots using hItln2 specific antibody (Bevins Lab, 1:5000) and goat-anti-rabbit HRP (1:10,000, Jackson, Cat# AB_2313567)[19].

Purification of StrepII-tagged lectins followed established procedures[12]. The harvested expression medium was subjected to avidin (IBA, Cat# 2-0204-015) treatment, resulting in a final concentration of 0.084 mg/mL, as per the IBA protocol. Protein was captured on Strep-Tactin Superflow High-Capacity resin (IBA, Cat# 2-1208-002), which was preequilibrated with HEPES/Ca buffer (20 mM HEPES, 10 mM $CaCl_2$, 150 mM NaCl, pH 7.4). Subsequently, the resin was washed twice with HEPES/EDTA buffer (20 mM HEPES, 1 mM EDTA, 150 mM NaCl, pH 7.4). The StrepII-tagged lectins were then eluted with 5 mM d-desthiobiotin (Sigma-Aldrich) in HEPES/EDTA buffer and concentrated using 10,000-molecular weight cutoff (MWCO) Vivaspin 6 centrifugal filter (Sartorius).

All proteins underwent buffer exchange to HEPES/EDTA for storage except for StrepII-hItln2 (UC/CD conditions). Purification of hItln2 in low pH and low salt (UC/CD condition) was identical, except that the purified protein was buffer exchanged into PIPES/EDTA buffer (10 mM PIPES, 25 mM NaCl, 1 mM EDTA, pH 5.5). Protein concentrations were determined by absorbance at 280 nM, with extinction coefficients and molecular weights calculated for the monomeric form of each protein (without the signal peptide) using the Protparam tool. StrepII-tagged hItln1 had $\varepsilon = 79,925$ cm$^{-1}$m$^{-1}$ and an estimated molecular mass of 34,224 Da (monomer). StrepII-tagged mItln2 had $\varepsilon = 69,830$ cm$^{-1}$m$^{-1}$ and an estimated molecular mass of 34,212 Da. StrepII-tagged hItln2 had $\varepsilon = 69,955$ cm$^{-1}$m$^{-1}$ and an estimated molecular mass of 34,436 Da (monomer).

## Detection of hItln2 in human fecal samples

Human fecal samples from three independent donors were kindly provided by the Alm lab (MIT)[13]. Fecal samples were supplied as homogenates in glycerol and stored at −80 °C. To evaluate the secretion of hItln2 in the intestinal lumen, human fecal homogenates were scraped from frozen glycerol stock and thawed on ice. The homogenate was centrifuged at $100 \times g$ for 1 min to pellet fibrous material. The supernatant was transferred to a new tube and pelleted at $3000 \times g$, 5 min. The pellet was washed twice with PBS, and then masses of pellets were measured. Pellets were suspended in PBS to achieve a concentration of 40 mg/mL. 20 mL of each sample was mixed with 6x Laemmli buffer containing DTT (Dithiothreitol), boiled at 95 °C, and run on 4–15% TGX Stain-Free gel (Bio-Rad). Total protein was visualized using stain-free settings on ChemiDoc XP (BioRad). Proteins were transferred to PVDF, blocked in 5% milk in TBST, and blotted for hItln2 protein using an anti-hItln2 antibody (Bevins Group, 1:5000) and Goat-anti-rabbit HRP (1:10,000, Jackson, Cat# AB_2313567)[19].

## Treatment of mItln2 with PNGase F

To examine N-glycosylation, 20 μg of recombinant mItln2 was subjected to PNGase F (NEB, Cat# P0704S) treatment according to the manufacturer's recommendations. Following glycosidase treatment, samples were mixed with 6x Laemmli buffer containing dithiothreitol (DTT), boiled at 95 °C, and run on 4–15% TGX stain-free gel (Bio-Rad) with molecular weight standards (Bio-Rad). Gel was imaged on ChemiDoc MP Imaging system using stain-free imaging settings.

## Circular dichroism (CD) analysis

Recombinant mItln2 was purified for CD analysis as described above, except all washes, elution, and buffer exchanges were performed with PBS buffer. CD spectra from 180 nm to 260 nm with 1 nm wavelength steps were collected on a Jasco H-1500 (Jasco Inc) at 25 °C. Measurements were conducted with 500 μg/mL of mItln2 (in PBS buffer) using a 1 mm cuvette, with PBS buffer serving as the blank. Averaging time was 5 s, settling time was 0.33 s, and readings were taken in triplicate. Thermal denaturation steps were run from 25 °C to 95 °C with 5 °C temperature steps. Scans were averaged, corrected by subtracting the blank, and plotted in GraphPad Prism to calculate melting temperature and visualize spectra. The calculated mean molar ellipticity (θ) is plotted as a function of wavelength (nm) for the protein.

## Differential scanning fluorometry (DSF) and dynamic light scattering (DLS)

Particle light scattering and change in intrinsic fluorescence intensity of recombinant lectins at 330 nm and 350 nM in response to temperature changes were measured using a NanoTemper Prometheus NT.48 instrument. Capillaries were filled with -10 μL of recombinant lectins (in HEPES/EDTA buffer or PIPES/EDTA buffer), placed into the sample holder, and the temperature was increased from 25 °C to 90 °C. Typical lectin stock concentrations for testing were 5–20 μM. The fluorescence intensity ratio at 350 nm and 330 nm was plotted as a function of temperature, and its first derivative was calculated using the manufacturer's software. The melting temperatures were calculated as the inflection point of the ratio curve. DLS was measured at 25 °C prior to melting temperature analysis. The hydrodynamic radii were determined using buffer-only capillaries as a baseline. The DLS and DSF experiments were conducted using two replicates for each sample.

## Chemical cross-linking of proteins

For mItln2 cross-linking, 1 mg/mL of mItln2 (in HEPES/EDTA buffer) was combined with BS3 (ThermoFisher, Cat# 21580) cross-linker in HEPES/EDTA buffer to achieve different final concentrations (0.1 mM,

0.25 mM, 0.5 mM, 1.25 mM, 2.5 mM, and 5 mM). Cross-linking was performed at RT for 30 min. Subsequently, protein mixtures were denatured by adding an SDS loading buffer with DTT. Samples were then heated at 95 °C for 5 min, separated by SDS-PAGE, and imaged with ChemiDoc MP Imaging system using SYPRO Ruby protein stain or TGX Stain Free technology.

## Enzyme-linked lectin assay (ELLA)

To assay mItln2 binding to different carbohydrates, a 96-well MaxiSorp ELISA plate (ThermoFisher, Cat# 442404) was coated with 0.5 μg of streptavidin (Agilent, Cat# SA26) in PBS per well. Subsequently, the wells were loaded with 5 μM of biotinylated carbohydrates (in PBS) for 1 h at RT. After blocking with 5% BSA in HEPES/Ca buffer, the plate was incubated with various concentrations of recombinant lectins in HEPES/Ca/BSA/T buffer (20 mM HEPES, 10 mM CaCl$_2$, 150 mM NaCl, 0.1% BSA, 0.1% Tween-20) for 2 h at RT. Following washes with the HEPES/Ca/BSA/T buffer, the wells were treated with anti-StrepMAB-Classic HRP conjugate antibody (1:10,000, IBA, Cat#2-1509-001) in HEPES/Ca/BSA/T buffer for 2 h at RT. Then, the plates were washed with HEPES/Ca/BSA/T buffer, and the bound StrepII-tagged lectins were detected colorimetrically using 1-Step Ultra TMB-ELISA (ThermoFisher). The reaction was quenched by adding an equal volume of 2 M sulfuric acid. Plates were read at 450 nm on Molecular Devices SpectraMax5. Data were analyzed using GraphPad Prism.

## Mucin dot blot

Purified porcine MUC2, MUC5AC, and MUC5B were generously provided by the Ribbeck Lab at MIT. Partially purified gastric mucus was purchased from Sigma (Cat# M1778). Mucins were hydrated overnight in deionized water, 4 °C, with rotation to achieve a stock concentration of 10 mg/mL. Mucins and gastric mucus were serially diluted in 20 mM HEPES (pH 7.4) and 150 mM NaCl buffer to generate working stocks for downstream assays. For individual mucins, 1 μg of hydrated mucins were spotted onto nitrocellulose and allowed to dry. After blocking with 5% BSA in TBS-T for 1 h at RT, 0.5–1 μM recombinant lectins in HEPES/Ca/BSA/T buffer or HEPES/EDTA/BSA/T buffer (20 mM HEPES, 1 mM EDTA, 150 mM NaCl, 0.1% BSA, 0.1% Tween-20) were applied to blot overnight at 4 °C. To test the glycan specificity of lectin–mucin interactions, the blots were incubated with the lectins in combination with 10 mM galactose in HEPES/Ca/BSA/T buffer. For UC/CD conditions, the blot was incubated with hItln2 in PIPES/Ca/BSA/T buffer (10 mM PIPES, pH 5.5, 25 mM NaCl, 10 mM CaCl$_2$, 0.1% BSA, 0.1% Tween-20) or PIPES/EDTA/BSA/T buffer (10 mM PIPES, pH 5.5, 25 mM NaCl, 1 mM EDTA, 0.1% BSA, 0.1% Tween-20). Blots were washed in TBS-T and treated with anti-StrepMAB-Classic HRP conjugate antibody (1:10,000, IBA, Cat# 2-1509-001) in the appropriate buffer. Subsequently, the blot was washed three times with TBS-T and developed in the ChemiDoc MP Imaging system using ECL Prime reagent (Amersham).

For gastric mucus binding, serial dilutions of mucus were spotted on nitrocellulose and blocked as described above. Membranes were incubated with 0.5 μM strepII-mItln2 in HEPES/Ca/BSA/T alone or in combination with 10 mM glucose or 10 mM galactose. Blots were incubated for 1 h at RT or overnight at 4 °C, processed as described above, and imaged using autoradiography film (Cole-Palmer) and developing reagents (Kodak).

## Mucin and glycopolymer agglutination assay

To test the agglutination effect of mItln2, we adapted an assay from Järvå et al.[41]. 1% (w/v) mucins or glycopolymer were first diluted to 0.35% (w/v) in HEPES/Ca or PIPES/ Ca$^{2+}$ buffer. 1 μL mucin or glycopolymer was aliquoted into each well of a 384-well plate (black/clear bottom). Using a multichannel pipette, 34 μL of increasing concentrations of Itln2 in HEPES/Ca or PIPES/Ca buffer or buffer alone was added simultaneously to wells and briefly mixed to achieve a final concentration of 0.01% (w/v) mucin. The plate was immediately

monitored for changes in absorbance at 405 nm (A405nm) using Molecular Devices SpectraMax5 via a kinetic assay, taking reads every 10 s for 30 min at RT. Triplicates were averaged and plotted in GraphPad Prism. Data is displayed as changes in A405nm as a function of time.

## Imaging of lectin-mediated mucin agglutination

Purified porcine MUC2, MUC5AC, and MUC5B were hydrated in deionized water overnight at 4 °C, with rotation to prepare stock concentration of 10 mg/mL. Mucins were labeled with AlexaFluor 568 NHS ester (ThermoFisher, Cat# A20006) as per the manufacturer's recommendations. Labeled mucins were pelleted at 10,000 × g for 10 min and washed thrice with excess PBS to remove unreacted fluorophore. Mucins were resuspended to a working concentration of 1% (w/v) in HEPES/Ca or PIPES/Ca buffer. Mucins were plated in a lysine-coated 96-well imaging plate, which was followed by the addition of solutions containing 5 μM lectin, StrepMAB-Classic DY649 (IBA, Cat# 2-1569-050, 1:250) in HEPES/Ca/BSA/T or PIPES/Ca/BSA/T. Mucins treated with only buffers were used as a control. The final mucin concentration in the assay was 0.01% (w/v). Following treatment with lectins, the plate was centrifuged briefly at 1000 × g for 1 min and immediately imaged on Molecular Devices IXM HC confocal microscope. Images were acquired at the start of the experiment and every 15 min after that for a total of 2 h. Images were analyzed in Fiji, where settings required for the brightest images were applied to all samples. Signal appears lower for untreated mucins for this reason.

## Synthesis of biotin-carbohydrates and glycopolymers

Synthesis of biotinylated carbohydrates (β-Galf-biotin and β-Galp-biotin) and 60% lactose-functionalized trans-poly(norbornene) polymer, which were synthesized in-house, has been described previously[12,44]. Other biotinylated carbohydrates (α-Galp, β-LacNAc, β-GlcNAc, α-Neu5Ac, β-6-SO$_3$ LacNAc) were purchased from GlycoTech.

## Biolayer interferometry (BLI)

Lectin binding to biotinylated carbohydrates was assessed using the OctetRed BLI instrument (ForteBio). Biotinylated carbohydrate was loaded onto streptavidin biosensor (ForteBio) for 120 s as a 5 μM solution in PBS buffer. The sensor was subsequently washed in PBS for 60 s, and a baseline was established in HEPES/Ca/BSA/T or PIPES/Ca/BSA/T buffer for 120 s. Various concentration of recombinant lectins (in HEPES/Ca/BSA/T or PIPES/Ca/BSA/T buffer) was then associated for 600 s followed by dissociation in HEPES/Ca/BSA/T or PIPES/Ca/BSA/T buffer for 600 s. The shake rate was maintained at 1000 rpm throughout the experiment, and the binding assays were performed at 30 °C. Data were normalized through background subtraction using the response from biotin-loaded streptavidin. The results were analyzed using GraphPad Prism.

To determine the dissociation constant for lectin with biotinylated carbohydrates, we conducted the BLI experiments with various lectin concentrations using the same protocol as above. The response at equilibrium was plotted against the concentration of lectin, and the resulting curve was fit to a one-site total non-linear regression equation to determine the equilibrium dissociation constant (GraphPad Prism).

## Assay of Itln2 on microbial and mammalian glycan array

Lectin binding to mammalian glycans was assessed using the Consortium for Functional Glycomics (CFG) version 5.5 microarray and RayBiotech slides (Cat# GA-Glycan-300-1). Binding to microbial glycans was assessed using the CFG microbial glycan microarray version 2 (MGM). Each glycan sample on the microarray had six replicates. The microarray slides were first rehydrated for 5 min in HEPES/Ca/BSA/T buffer. Subsequently, the slides were incubated with 25 μg/mL of StrepII-tagged mItln2 in HEPES/Ca/BSA/T buffer for 1 h at RT. The slides were then washed with HEPES/Ca/BSA/T buffer and incubated

with StrepMAB-Classic DY-549 (1:250, IBA, Cat# 2-1566-050) in HEPES/Ca/BSA/T buffer for 1 h at RT. Afterward, the array underwent washing steps using HEPES/Ca/BSA/T buffer, HEPES/Ca buffer, and deionized water. Finally, the bound lectin in the microarray was detected using a fluorescent scanner. The results are presented as relative fluorescence units obtained by averaging the background-subtracted signals for the four replicate spots (after excluding the highest and lowest values from the six replicates), with error bars representing the SD of the averaged values. The microbial glycan array was interpreted using the key from the original publication[39]. Of note, hItln2 was assayed for binding on the mammalian glycan arrays (CFG v5.5) but, for unclear reasons, the lectin did not display binding. Alternative approaches (dot blot, BLI, and microscopy) showed that mItln2 and hItln2 have similar binding patterns.

The differences in fluorescence intensities between experiments reflect different time points of data collection and differences in the glycan array slides and fluorescence scanners used. The mItln2 CFG mammalian glycan array data (Fig. 2B) was collected in 2016 using ProScanArray Scanner (Perkin Elmer) instrument at the core facility. The microbial glycan array data (Fig. 2B) and the RayBiotech mammalian array data (Supplementary Fig. 2B) for mItln2 were collected in 2021 in-house using a Genepix 4400A scanner (Molecular Devices). Similarly, the hItln2 microbial glycan array data (Fig. 5E) was collected in 2024 using a Genepix 4400A scanner (Molecular Devices) instrument at the core facility. During the 2024 run, the core facility noted that the scanner in use during that time had a maximum lower relative fluorescence unit (RFU) of approximately 4000 compared to 60,000 on the previous scanner, resulting in lower RFU values. Moreover, in 2024, the core facility also changed protocols and began regenerating and reusing slides; the long-term effects on fluorescence intensity are unclear[69].

## Assay for lectin binding to fecal samples

Freshly collected murine fecal samples were homogenized in PBS. Human fecal homogenates were scraped from frozen glycerol stock and thawed on ice. The homogenate was centrifuged at $100 \times g$ for 1 min to pellet fibrous material. The supernatant was transferred to a new tube and pelleted at $3000 \times g$, 5 min. The pellet was washed twice with PBS, and the optical density at 600 nm ($OD_{600}$) was measured. To test Itln2 binding to fecal microbiota, the homogenate was diluted to $OD_{600}$ of 0.2 and treated with 0.6 µM of StrepII-tagged mItln2, StrepMAB-Classic DY549 antibody (1:250, IBA, Cat# 2-1566-050), or 1 µM StrepII-hItln2, StreptactinXT-DY549 (1:250, IBA, Cat# 2-1565-050) and SYTO BC (1:1000, ThermoFisher, Cat# S34855) in HEPES/Ca/BSA/T buffer for 2 h at 4 °C. To test $Ca^{2+}$-dependent binding of Itln2, staining was carried out in HEPES/EDTA/BSA/T buffer and used as a control. For the hItln2 in UC/CD condition, the homogenate was treated with 1 µM StrepII-hItln2, StreptactinXT-DY549 (1:250, IBA, Cat# 2-1565-050) and SYTO BC (1:1000, ThermoFisher, Cat# S34855) in PIPES/Ca/BSA/T buffer or PIPES/EDTA/BSA/T buffer for 2 h at 4 °C. Untreated fecal samples and the fecal samples treated solely with StrepMAB-Classic DY549 antibody, StrepTactinXT-549, or SYTO BC were used as additional controls. Following staining, the cells were analyzed on LSRII, LSR Fortessa HTS flow cytometer or FACS Symphony flow cytometer. Flow data were analyzed with FlowJo.

For the microscopy experiment, the fecal samples were processed identically to the flow experiment described earlier. Following staining, the samples were transferred to a lysine-coated 96-well black/clear bottom Plate (ThermoFisher, Cat# 165305) and centrifuged at $500 \times g$ for 1 min. Imaging was performed using the RPI Spinning Disk Confocal microscope or Molecular Devices IXM HC confocal microscope. Image analysis was performed using Fiji.

## Assay for lectin binding to microbial isolates

Bacterial isolates *Bacillus paramycoides* (RJX001), *Bacillus paramycoides* (RJX007), *L. reuteri* (RJX004), *L. murinus* (RJX002), and *L.*

*johnsonii* (RJX009) were isolated from mouse stool by Xavier Lab (Broad Institute). *Lactobacillus reuteri* (DSM 20016) and *Escherichia coli* DH5α were obtained from Prof. Federico Rey, courtesy of Dr. Robert Kerby (UW-Madison) and NEB, respectively. *B. paramycoides* (RJX001), *B. paramycoides* (RJX007), *L. reuteri* (DSM 20016 and RJX004), *L. murinus* (RJX002), and *L. johnsonii* (RJX009) were grown anaerobically in a chamber with AnaeroGen sachet (ThermoFisher) at 37 °C for 16 to 24 h without shaking. *B. paramycoides* (RJX001 and RJX007) were grown in CHG medium [brain heart infusion (37 gL$^{-1}$, BD) supplemented with 1% vitamin K1-hemin (ATCC), D-(+)-cellobiose (1 gL$^{-1}$, Sigma-Aldrich), D-(+)-fructose (1 gL$^{-1}$, Sigma-Aldrich), D-(+)-maltose (1 gL$^{-1}$, Sigma-Aldrich), and L-(+)-cysteine (1 gL$^{-1}$, Sigma-Aldrich). *L. reuteri* (DSM 20016 and RJX004), *L. murinus* (RJX002), and *L. johnsonii* (RJX009) were grown in MRS medium (BD). *E. coli* was grown in Luria Broth (LB) at 37 °C for 16 to 24 h with shaking at 200 rpm.

To test lectin binding, cells were revived from glycerol stock in their respective buffers and grown overnight in appropriate aerobic or anaerobic conditions. Cultures were harvested by centrifugation and washed with PBS twice before $OD_{600}$ measurement. Staining was performed at $OD_{600}$ of 0.2 for all samples. For staining, cells were treated with various concentrations of StrepII-tagged lectins, StrepMAB-Classic DY549 antibody (1:250, IBA, Cat# 2-1566-050) or StreptactinXT-DY549 (1:250, IBA, Cat# 2-1565-050), and SYTO BC (1:1000, ThermoFisher, Cat# S34855) in HEPES/Ca/BSA/T buffer for 2 h at 4 °C. To test the $Ca^{2+}$-dependent binding of lectins, staining was carried out in HEPES/EDTA/BSA/T buffer and used as a control. For UC/CD conditions, cells were treated with lectins in PIPES/Ca/BSA/T or PIPES/EDTA/BSA/T buffers. Unstained bacteria, bacteria treated only with StrepMAB-Classic DY549 antibody or StreptactinXT-DY549, and bacteria treated only with SYTO BC were used as additional controls. Following staining, the cells were diluted five-fold and analyzed on LSR Fortessa HTS flow cytometer or FACS Symphony flow cytometer. Data were analyzed with FlowJo.

For the microscopy experiment, the microbial isolates were processed identically to the flow experiment described earlier. Following staining, the samples were transferred to a lysine-coated 96-well black/clear bottom Plate (ThermoFisher, Cat# 165305) and centrifuged at $500 \times g$ for 1 min. Imaging was performed using Molecular Devices ImageXpress Micro Confocal Microscope, RPI spinning disc confocal, or Olympus FV1200 Laser Scanning Confocal Microscope. Image analysis was performed using Fiji.

## Assay for lectin binding to pathogenic isolates (*S. aureus*, *K. pneumoniae*, and *S. pneumoniae*)

*Staphylococcus aureus* USA300Lac (MRSA) and *Klebsiella pneumoniae* UCI60 were generously shared by Xavier and Hung groups (Broad Institute), respectively. Cells were grown in suspension of LB at 37 °C for 16 to 24 h with shaking at 200 rpm. *S. pneumoniae* (Klein) Chester serotypes 70 (ATCC10370) were obtained from ATCC. Cells were grown in suspension of Todd Hewitt broth (BD) with 0.5% yeast (BD) without shaking at 37 °C under 5% $CO_2$ for 24 h.

To test lectin binding, cells were harvested by centrifugation, washed with PBS, and fixed in 4% formaldehyde in PBS for 1 h at RT. Fixation was neutralized with Tris buffer (pH 7.4) at a working concentration of 50 mM. Of note, previous studies have compared lectin binding to live versus formaldehyde-fixed bacteria and found that fixation had minimal impact on lectin binding patterns, suggesting that the surface glycans are preserved during fixation[53]. After washing cells with PBS, $OD_{600}$ was measured, and staining was performed at $OD_{600}$ of 0.2. For staining, fixed cells were treated with various concentrations of StrepII-tagged lectins, StrepMAB-Classic DY549 antibody (1:250, IBA, Cat# 2-1566-050) or StreptactinXT-DY549 (1:250, IBA, Cat# 2-1565-050), and SYTO BC (1:1000, ThermoFisher, Cat# S34855) in HEPES/Ca/BSA/T buffer for 2 h at 4 °C. To test $Ca^{2+}$-dependent binding of lectins, staining was carried out in HEPES/EDTA/BSA/T buffer and

used as a control. For UC/CD conditions, cells were treated in PIPES/Ca/BSA/T or PIPES/EDTA/BSA/T buffers. Unstained bacteria, bacteria treated only with StrepMAB-Classic DY549 antibody or StreptactinXT-DY549, and bacteria treated only with SYTO BC were used as additional controls. Following staining, the cells were diluted five times without washing and analyzed on LSR Fortessa HTS flow cytometer or FACS Symphony flow cytometer. Data were analyzed with FlowJo.

For the microscopy experiment, the microbes were processed identically to the flow experiment as described earlier. Following staining, the fixed samples were transferred to a lysine-coated 96-well black/clear bottom Plate (ThermoFisher, Cat# 165305) and centrifuged at $500 \times g$ for 1 min. Imaging was performed using Molecular Devices ImageXpress Micro Confocal Microscope or Olympus FV1200 Laser Scanning Confocal Microscope. Image analysis was performed using Fiji.

### Colony-forming unit (CFU) Assays
*L. reuteri* (DSM 20016) was cultured anaerobically in MRS medium at 37 °C for 16 h without shaking. *B. paramycoides* RJX001 grown anaerobically in a chamber with AnaeroGen sachet (ThermoFisher) at 37 °C for 16 to 24 h without shaking. *S. aureus* USA300Lac (MRSA) was cultured in LB medium at 37 °C for 16 h with shaking at 200 rpm. Upon reaching the mid-logarithmic phase, the cells were harvested by centrifugation and washed with PBS before measuring $OD_{600}$. Subsequently, cells were diluted to $OD_{600}$ of 0.2 and treated with various concentrations of StrepII-tagged lectins in HEPES/Ca/BSA/T or PIPES/Ca/BSA/T buffer for 4 h (at 4 °C for *L. reuteri* and *S. aureus* and 37 °C for *B. paramycoides*). Following incubation with lectins, cells were serially diluted in sterile HEPES/Ca ($10^1$–$10^5$) or PIPES/Ca buffer and plated on appropriate agar plates (MRS for *L. reuteri*, CHG for *B. paramycoides*, or LB for *S. aureus*) and incubated overnight at 37 °C. *L. reuteri* and *B. paramycoides* plates were grown in Oxoid AnaeroJar 2.5 L anaerobic chamber with AnaeroGen 2.5 L sachet (ThermoFisher, AN0025A). Surviving bacterial colonies were counted, and CFU/mL was calculated. Data is displayed as a percentage relative to the control sample (cells treated with HEPES/Ca/BSA/T buffer or PIPES/Ca/BSA/T buffer without lectins).

### Suspension-culture growth assay for *K. pneumoniae*
For lectin-mediated growth inhibition assays, *K. pneumoniae* UCI60 was cultured directly from glycerol stock in LB media overnight at 37 °C with shaking. Cells were back-diluted in fresh LB and grown to mid-log phase ($OD_{600}$ of 0.4–0.7). Cells were harvested by centrifugation, washed with PBS, and the $OD_{600}$ was measured. Cells were then diluted in M9 growth media (1x M9 salts, 0.1% glucose, 100 μM $CaCl_2$, 2 mM $MgSO_4$) to achieve a 2x stock of cells ($OD_{600} = 0.001$). 2x cells were mixed with 2x stocks of increasing concentration of lectins in M9 media in a final volume of 50 μL. The plate was covered with BreathEasy gas-permeable film (Diversified Biotech, BEM-1) and allowed to grow overnight, 18 h, at 37 °C in SpectraMax M5 plate reader. $OD_{600}$ nm reads were taken every 15 min. Triplicate samples were averaged, and the mean $OD_{600}$ nm was plotted as a function of time (GraphPad Prism).

To determine the minimum inhibitory concentration (MIC) for lectins, the $OD_{600}$ measurements at 960 min for different lectin-treated conditions were normalized to $OD_{600}$ measurement using the no-lectin treatment conditions. The resulting normalized $OD_{600}$ was used to determine % of bacterial growth and fitted to the modified Gompertz functions to obtain MIC value as described before[70].

### Suspension-culture growth assay for *Lactobacilli*
As *Lactobacilli* do not grow in M9 minimal media, an alternate growth recovery assay was designed to assay lectin-mediated growth assay. *L. reuteri* (DSM 20016 and RJX004), *L. murinus* (RJX002), and *L. johnsonii* (RJX009) were cultured anaerobically in MRS medium at 37 °C for 16 h

without shaking. Cells were harvested by centrifugation, washed with PBS, and $OD_{600}$ measured. Subsequently, the cells were diluted to $OD_{600}$ of 0.2 and treated with various concentrations of StrepII-tagged lectins in HEPES/Ca/BSA/T buffer in a sterile 96-well round-bottom plate. Following treatment, cells were diluted five-fold in MRS broth, sealed with film, and allowed to recover overnight on SpectraMax M5 plate reader, 37 °C without shaking, taking $OD_{600}$ reads every 15 min. The growth assay was run in triplicate, and the data was averaged. Data was displayed as mean $OD_{600}$ changes plotted as a function of time.

### Time-lapse microscopy of *L. reuteri*
*L. reuteri* (DSM 20016) was cultured anaerobically in MRS medium at 37 °C for 16 h without shaking. To assess the impact of lectin on *L. reuteri*, the bacterial cells were harvested by centrifugation and washed with PBS before $OD_{600}$ measurement. Subsequently, the cells were diluted to $OD_{600}$ of 0.2 and treated with various concentrations of StrepII-tagged lectins in HEPES/Ca/BSA/T buffer in a lysine-coated 96-well black/clear bottom Plate (ThermoFisher, Cat# 165305). Imaging was performed using a 40x water immersion objective in Molecular Devices ImageXpress Micro Confocal Microscope at 10 min intervals over 6 h. Images were processed in Fiji.

### Assay for propidium iodide uptake in *B. paramycoides*
*B. paramycoides* RJX001, grown anaerobically in a chamber with AnaeroGen sachet (ThermoFisher) at 37 °C for 16 to 24 h without shaking, was harvested at mid-logarithmic phase by centrifugation and washed with PBS before measuring $OD_{600}$. Subsequently, cells were diluted to $OD_{600}$ of 0.2 and treated with various concentrations of StrepII-tagged mItln2 in HEPES/Ca/BSA/T or HEPES/EDTA/BSA/T buffer for 4 h at 37 °C. Following incubation with lectins, cells were treated with propidium iodide (ThermoFisher, Cat# P3566) for 20 min and analyzed using FACS Symphony flow cytometer.

### Assay for lectin binding and propidium iodide uptake in mammalian cells
HEK-293T cells were maintained in DMEM medium (ThermoFisher, Cat# 11995) supplemented with 10% heat-inactivated FBS, 1x penicillin-streptomycin, 1x L-glutamine, and 1x non-essential amino acids. To assay lectin binding, the HEK-293T cells were dissociated with EDTA and washed with DPBS. Subsequently, cells (100 K/staining condition) were washed with HEPES/Ca or HEPES/EDTA buffer and then incubated with various concentrations of StrepII-tagged lectins and StrepMAB-Classic DY-549 (1:250, IBA, Cat# 2-1566-050) in HEPES/Ca/BSA/T or HEPES/EDTA/BSA/buffer for 3 h at 4 °C. Following incubation with lectins, the cells were treated with 1 μL of Propidium iodide (ThermoFisher, Cat# P3566) for 20 min at 4 °C. The cells were then washed with the respective buffer twice and analyzed using an Attune flow cytometer.

### Liposome preparation
Palmitoyl oleoyl phosphatidylcholine (POPC) was obtained as a 25 mg/mL solution in $CHCl_3$ from Avanti Polar Lipids (Cat# 850457 C). 1,2-dioleoyl-sn-glycero-3-phosphoethanolamine-N-dibenzocyclooctyl (18:1 DBCO PE) was obtained as a 10 mg/mL solution in $CHCl_3$ from Avanti Polar Lipids (Cat# 870129C). The lipid solutions were stored at −20 °C and used within 3 months. Cholesterol was obtained as a white solid from Sigma-Aldrich (Cat# C8667).

Prior to preparing a lipid film, the lipid solutions were warmed to ambient temperature for at least 30 min to prevent condensation from contaminating the solution and degrading the lipid film. 330 μL of POPC, 75 μL of 18:1 DBCO PE, and 100 μL of a 4 mg/mL cholesterol solution in $CHCl_3$ were added to a glass scintillation vial. The solvent was removed with a gentle stream of nitrogen, and the resulting lipid film was stored under a high vacuum for a minimum of 12 h prior to use.

The dried lipid film was rehydrated with 1 mL of 150 mM NaCl, 20 mM HEPES, 10 mM CaCl₂, pH 7.4 containing 200 µg/mL Texas Red-dextran conjugate MW 3000 Da (Thermo Fisher, Cat# D3329), and vortexed vigorously for approximately 3 min to form a suspension of multilamellar vesicles (MLVs). To obtain a sufficient quantity of large unilamellar vesicles (LUVs), at least three independent lipid film preparations were pooled together for the subsequent formation of LUVs. The resulting lipid suspension was pulled into a Hamilton (Reno, NV) 1 mL gastight syringe and the syringe was placed in an Avanti Polar Lipids Mini-Extruder (610000). The lipid solution was then passed through a 200 mm pore size hydrophilic polycarbonate filter (Avanti Polar 610006) 21 times, the newly LUV suspension being collected in the syringe that did not contain the original suspension of MLVs to prevent the carryover of MLVs into the LUV solution. The newly formed LUVs were purified via size-exclusion chromatography (SEC) using Sephadex G-50 (Sigma-Aldrich G50150) and functionalized with 5 mM β-Lactose-PEG3-azide (Sigma-Aldrich SMB00404-25MG) with shaking at 4 °C overnight via a strain-promoted azido-alkynyl cycloaddition (SPAAC). The lactose-functionalized liposomes were then purified by SEC to remove unreacted glycan ligands.

### Liposome membrane disruption assay

Liposomes were diluted to OD 0.1 in 150 mM KCl, 20 mM HEPES, 10 mM CaCl₂, pH 7.4 buffer. Absorbance (405 nm) was measured with a SpectraMax microplate reader (Molecular Devices). 50 µL of the liposome suspension was added to a 96-well plate, and initial measurement was recorded. Then, 25 µL of mItln2 or vehicle (buffer only) was added to each well, and absorbance was continuously read for 45 min, with measurements taken every 30 s.

For the microscopy experiment to visualize liposomal structural integrity, following measurement of absorbance over time following lectin addition, samples were imaged on an RPI Spinning Disc Confocal. Image analysis was performed using Fiji.

### STAT6 motif analysis

The promoter sequence for mItln2 in BALB/c mouse genome (Accession: GCA_921997145.2, https://www.ncbi.nlm.nih.gov/datasets/genome/GCA_921997145.1/) was retrieved from Ensemble (projects.ensembl.org/mouse_genomes/). The promoter sequence for hItln2 was retrieved from GRCh38/hg38 assembly (genome.ucsc.edu/). Then, the promoter regions (−2000 base pairs to −1 base pair relative to the transcription start site) of mItln2 and hItln2 were scanned for the presence of STAT6 motif [obtained from JASPAR (jaspar.elixir.no/)] using FIMO version 5.5.8 with default parameters[34]. The sequences that matched the STAT6 motif, along with their log-odds score (score for the match), p-value, and q-value (false-discovery rate) are reported.

### Statistical analysis

Statistical analysis of the data was performed using GraphPad Prism 9. Unpaired Student's t-test was used when comparing two independent samples. One-way ANOVA with Dunnett's multiple comparisons test was used when comparing three or more samples. Three biological replicates were used, except when indicated otherwise. ns: not significant, $*p < 0.05$, $**p < 0.01$, $***p < 0.001$, $****p < 0.0001$.

### Software

Protein structures and overlays were generated with PyMol (The PyMol Molecular Graphics System, Version 3.0 Schrodinger, LLC). Putative structures of mItln2 and hItln2 were generated with AlphaFold3[38]. Flow cytometry data was analyzed using FlowJo (FlowJo™ v10.8 for Mac Softwarejm). Imaging data was analyzed using Fiji[71]. GraphPad Prism version 10.1.1 for Mac was used for data analysis and graph generation (GraphPad Software, Boston, MA, USA, www.graphpad.com). Sequencing data was analyzed using SnapGene (www.snapgene.com). Carbohydrate structures were generated using ChemDraw version

20.1 for Mac (Revvity Signals Software). References were cited using Zotero version 6.0.37 (www.zotero.org). Schematics in figures were generated with BioRender.

### Reporting summary

Further information on research design is available in the Nature Portfolio Reporting Summary linked to this article.

## Data availability

The glycan array data for mItln2 and hItln2 are provided in Supplementary Data sets 1 and 2 (the corresponding legends are provided in Supplementary Dataset 3). Data to generate graphs in the main and supplemental text are available in the Source data file. Any additional information required to reanalyze the data reported in this paper is available from the lead contact upon request. Source data are provided with this paper.

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

## Acknowledgements

The authors thank the MIT Biophysical Instrumentation Facility, Drs. Bradley Turner and Mike Wuo, for assistance with circular dichroism and biolayer interferometry. We thank Drs. Robert Kerby and Federico Rey for *L. reuteri* DSM20016 strain, Dr. Edward Hollox for discussions on the B6.C-Itln1-6 mouse model, and Prof. Eric Alm for generously providing human fecal samples. The authors thank the Koch Institute's Robert A. Swanson (1969) Biotechnology Center for microscopy and flow cytometry core facilities. The research reported in this publication was supported by NIH Glycoscience Common Fund U01 CA231079 (L.L.K.), NIAID R01 AI055258 (L.L.K.), U01 AI125926 (C.L.B.), R37 AI32738 (C.L.B.), R21 AI176012 (C.L.B.), NIGMS T32GM007753 (J.J.Y.), NIGMS T32GM144273 (J.J.Y.), NIDDK P30 DK043351 (R.J.X.), DK127171 (R.J.X.), AI172147 (R.J.X.), DK135492 (R.J.X.), NSF EF-2125118 (K.R.), and ICB-2024-BEM-10 (under award W911NF1920026) (K.R.). The content is solely the responsibility of the authors and does not necessarily represent the official views of the NIGMS or NIH. A.E.D. thanks Dr. Kittikhun Wangkanont and Richard Barnocki for helpful discussions.

## Author contributions

Conceptualization: A.E.D., D.S. (Deepsing Syangtan), E.B.N., C.L.B., and L.L.K.; methodology: A.E.D., D.S. (Deepsing Syangtan), E.B.N., R.S.C., A.L.P., D.S. (Dallis Sergio), C.D., J.W.A., C.E.B., M.K., and G.C.O.; investigation: A.E.D., D.S. (Deepsing Syangtan), E.B.N., R.S.C., A.L.P., J.J.Y., J.I., D.S. (Dallis Sergio), C.D., S.B., S.J., C.E., S.G., J.W.A., C.E.B., M.K., and G.C.O.; visualization: A.E.D., D.S. (Deepsing Syangtan), E.B.N., R.S.C., A.L.P.; funding acquisition: K.R., R.J.X., C.L.B., and L.L.K.; project administration: G.P., H.V., K.R., R.J.X., C.L.B., and L.L.K.; supervision: K.R., R.J.X., C.L.B., and L.L.K.; writing—original draft: A.E.D., D.S. (Deepsing Syangtan), E.B.N., C.L.B., and L.L.K.; writing—review & editing: A.E.D., D.S. (Deepsing Syangtan), E.B.N., R.S.C., A.L.P., S.G., R.X., C.L.B., and L.L.K.

## Competing interests

The authors declare no competing interests.
