## [Transparent Peer Review file · Nature Communications]

Intelectin-2 is a broad-spectrum antimicrobial lectin

Corresponding Author: Professor Laura Kiessling

Version 0:

Reviewer comments:

Reviewer #1

(Remarks to the Author)

In their manuscript entitled "Intelectin-2 is a broad-spectrum antimicrobial lectin", Dugan et al. report an in-depth functional characterization of murine and human intelectin-2 (Itln2). The authors provide evidence for an interaction of both mItln2 and hItln2 with different host mucin isoforms as well as a variety of gram-negative and gram-positive bacteria, thus suggesting a role for Itln2 in microbial recognition. Moreover, they convincingly show direct antimicrobial effects of Itln2, thereby indicating a relevance for Itln2 in mucosal defense. In summary, the manuscript presents novel and important mechanistic insights into the role of intelectins in host-microbe interactions.

Overall, the present manuscript contains a wealth of experimental data that allow for elucidating the glycan preferences of both mItln2 and hItln2. It builds on previous studies that reported an upregulated expression of mouse Itln2 during nematode infection and postulated a role for Itln2 in inflammatory processes. By using an innovative combination of biophysical techniques, glycan arrays, biochemistry, and microbiology methods, the authors demonstrate a dual role of Itln2 for host protection, i.e., via binding to host-derived mucins on the one hand and recognition and killing of bacteria on the other hand. The manuscript is well written and of interest to a wide readership.

However, specific points need to be addressed to justify and/or support some conclusions drawn by the authors (see "Specific points" below).

Specific points:

- 1.) Figure 1: Using enteroids from a congenic C57BL/6 mouse model encoding the complete Itln1-6 locus, the authors show an induced Itln2 expression upon IL4 and IL-13 stimulation, according to previous studies reporting Itln2 induction during Th2 responses. In this assay, Th1 cytokines and Th1-inducing inflammatory stimuli should be included to assess the specificity of the Itln2 upregulation by Th2 cytokines. Moreover, it will be interesting to see whether Itln2 is also induced upon stimulation of the enteroids with killed bacteria and/or bacteria-derived PAMPs.
- 2.) Figures 2 and 5; Supplementary Tables 2 and 3: The authors determine the glycan specificity of mItln2 and hItln2 using different glycan arrays. They observe some hits for microbial glycans as well as mammalian glycans. However, the glycan array results require further explanation. For instance, it looks as if the RayBiotech mammalian glycan array yielded several (potentially unspecific?) hits, which seems to be in partial contrast to the CFG mammalian glycan array. Moreover, the fluorescence intensities are quite low in some instances (e.g., Figure 5E). It will be helpful and facilitate readability if the authors discuss the sometimes counterintuitive glycan array results in more detail and give further explanations.
- 3.) Figures 2E, 5G (mucin binding of mItln2 and hItln2): The authors show a direct binding of mucins to mItln2 and hItln2 and convincingly confirm these interactions by different methods, including dot blot, fluorescence microscopy, and spectroscopic assays. However, how specific are the interactions of mItln2 with the different mucin isoforms? Are there any mucin isoforms that are not recognized by mItln2? Can the authors provide an explanation for the differential binding of hItln2 (e.g., why MUC5B is not bound by hItln2)?
- 4.) Supplementary Figure 6: To optimize hItln2 expression, the authors mutated some amino acid residues for an improved solubility and reduced aggregation. Could these modifications have affected the binding properties of hItln2?
- 5.) According to the Methods section, binding studies to microbial isolates were done with live bacteria, while lectin binding to pathogenic isolates was determined using formaldehyde-fixed bacteria. Did the fixation procedure affect the binding

behavior of the bacteria (did the authors compare Itln2 binding to live vs. fixed bacteria)?

6.) Titration of Itln2 for determination of antimicrobial activity: In some cases, the authors used different Itln2 concentrations for bacteria binding and killing assays, which renders comparisons of antimicrobial activities of one Itln2 against different bacteria species and/or among mlItln2 and hlItln2 difficult. This should be made consistent by titration of both lectins over a wider concentration range. For instance, in Figure 4B, it looks as if only the highest concentration of 5 μ M mlItln2 is active against *L. reuteri* (i.e. no activity up to 2.5 μ M), whereas in a different assay for microbicidal activity, even the lowest concentration of 1 μ M Itln2 exhibited a maximum antimicrobial effect. Thus, K_i values should be measured to determine the effective antimicrobial concentrations of mlItln2 and hlItln2.

7.) The authors found that hlItln2 retained binding to beta-galactopyranose and additional ligands at low pH and ionic strength (Figure 7). Were similar experiments performed with mlItln2 as well? EDTA addition did not abrogate the interaction of hlItln2 with mucins and microbes at low pH and low salt concentrations, which is an interesting but somewhat unexpected finding. The authors suggest an "alternate binding mode" (p.7) as a potential explanation. Does that mean that the orientation of the carbohydrate ligands in their interaction with Itln2 is expected to be completely different under these conditions? Does such an altered binding mode have an impact on the specificity of the recognized microbes (i.e., is bacterial recognition of hlItln2 altered, as compared with Supplementary Figure 7)?

Reviewer #2

(Remarks to the Author)

This is a fine manuscript describing an animal lectin for which no activity could be assigned previously. The extensive study brings molecular characterization, specificity study with some measurement of affinity, and functional role against pathogen infection. The manuscript reads very well, but, presumably because of space limitation, lacks some better description of the large amount of experimental data.

Major points

The oligomeric state of the proteins is quickly discussed and some more description of experimental data would be helpful. If mlItln2 is composed of a mixture of different oligomers, does this mean that only part of the sample is correctly associated? What about the effect on activity? As for hlItln2, it is stated that it occurs as a covalent trimer, but Supp. Fig 6e is more in evidence of a hexamer (Supp Fig 6d should be explicated). A Cys involved in oligomerization has been mutated, what is the effect on oligomerization?

The glycan array data are of high importance. They provide the basis of the lectin specificity that was unknown for intelectin2. Fig 2B should include identification (labeling) of the highly positive ligands. Some more discussion about the observation would be of interest. In the mammalian array, results pointing to N-glycans and to O-glycans should be discussed: biantennary glycan with LacNAc repeats are excellent ligands, specially those with 3 LacNAc repeats (540 and 567) and no binding is observed with only 2 repeats. Triantennary is even better. More numerous LacNAc repeats result in weaker binding. This could be discussed in term of lectin topology. Similarly the binding of terminal sulfated Gal is not that simple: it is efficient only when bound to 6SGlcNAc or 6SGlc by b1-4 linkage. Figure 2C is rather misleading: the two disaccharides at the bottom are not recognized by the lectin.

It is not clear why hlItln2 was not assayed on the mammalian array.

The excel files provided in supplemental data are not easy to understand (except for CFG mammalian). There is no identification of the polysaccharides/microbes in the microbial analysis, and only code names for the RayBiotech data. This glycan array is never mentioned in the main text (Supp Fig 2b is cited with no explanation) and the strong discrepancy between SFG glycan array and RayBiotech is not discussed.

3D-models are important for comparing intelectin-2 (mouse and human) with intelectin-1. In the present version, one model is obtained with Swissmodel and the other with Alpha-Fold3 without justification. The same procedure (either one or the other) should be used for both. The two methods also allows for modeling oligomeric state and it would be of interest to have a view of the modelled trimers. Alignment of sequences is provided only for 2 sequences in Supp Fig1a.. It would be useful to have a global alignment for mouse and human intelectins (1 and 2). Finally docking galactose would not be much work (for example with Autodock) and will rationalize the effect of sulfate (but of course could be kept for further study).

Reviewer #3

(Remarks to the Author)

This manuscript provides important new insights into mucosal immunity by characterizing the intelectin-2 (Itln2) family in both mice and humans. Building on the authors' prior work on intelectin-1 (Itln1), the study reveals distinct and complementary roles for Itln2 in host defense at mucosal surfaces. These lectins are expressed by epithelial cells in barrier tissues, and the current study focuses on their function in intestinal immunity.

The authors show that mouse Itln2 expression is induced by type 2 cytokines and that the protein exhibits antimicrobial activity via direct bacterial killing and membrane disruption. In contrast, human ITLN2 primarily agglutinates bacteria. Both forms can cross-link mucins, suggesting a dual role in antimicrobial defense and enhancing mucosal barrier integrity.

Importantly, the authors demonstrate that mltln2 binds terminal β -D-galactopyranose, a specificity distinct from mltln1 and hITLN1. This divergence is supported by structural data.

Overall, the data are quite robust and the conclusions are well-supported. The work makes a significant contribution to our understanding of the diversity of antimicrobial proteins (AMPs) at epithelial surfaces. That said, I have a few specific questions and suggestions that could further strengthen this already strong manuscript:

1. The microarray data indicate that intelectins also bind to mammalian glycans. More discussion is warranted on how carbohydrate structure enables discrimination between host and microbial targets. What prevents host cell targeting in vivo?
2. Several known AMPs show preferential activity against log-phase bacteria. Has the antimicrobial activity of mouse Itln2 been tested across different bacterial growth phases?
3. The liposome disruption assays use mammalian-type lipid compositions, yet these membranes are disrupted by mltln2. Can the authors clarify why they think that these proteins don't show toxicity to mammalian cell membranes?
4. What is the estimated concentration of mouse Itln2 in the intestine or enteroids? This would help contextualize the in vitro findings.
5. The section on pH sensitivity could be clarified. While the authors note that low pH and low salt are linked to inflammation, gut epithelial surfaces also experience these conditions physiologically. Where are Itln2 proteins expressed relative to such environments? Notably, their ability to function at pH 7.4 and 150 mM NaCl is unusual for AMPs and could be emphasized.
6. How do intelectins complement or differ from other AMPs? In other words, how are they solving a distinct "evolutionary challenge"? Is it just that they are induced by type 2 cytokines? Their broad bacterial specificity would suggest that they are not simply targeting a subset of bacteria that is "missed" by other AMPs.
7. Since type 2 cytokines promote goblet cell hyperplasia and mucus production, the co-induction of antimicrobial intelectins that also cross-link mucins is intriguing. This synergistic relationship could be highlighted more explicitly in the discussion.
8. Given the role of type 2 cytokines, are STAT6 binding motifs present in the promoters of the mltln2 and hITLN2 genes?
9. Do N-linked glycosylation sites on the recombinant proteins affect their oligomerization state or binding activity? Are any of the N-glycans capped with β -D-galactopyranose, potentially affecting specificity?

Reviewer #4

(Remarks to the Author)

Version 1:

Reviewer comments:

Reviewer #1

(Remarks to the Author)

In their manuscript entitled "Intelectin-2 is a broad-spectrum antimicrobial lectin", Dugan et al. report an in-depth functional characterization of murine and human intelectin-2 (Itln2). The authors provide evidence for an interaction of both mltln2 and hltln2 with different host mucin isoforms as well as a variety of gram-negative and gram-positive bacteria, thus suggesting a role for Itln2 in microbial recognition. Moreover, they convincingly show direct antimicrobial effects of Itln2, thereby indicating a relevance for Itln2 in mucosal defense. In summary, the manuscript presents novel mechanistic insights into the role of intelectins in host-microbe interactions.

Overall, the manuscript contains a wealth of experimental data that allow for elucidating the glycan preferences of both mltln2 and hltln2. The present manuscript builds on previous studies that reported an upregulated expression of mouse Itln2 during nematode infection and postulated a role for Itln2 in inflammatory processes. By using an innovative combination of biophysical techniques, glycan arrays, biochemistry, and microbiology methods, the authors demonstrate a dual role of Itln2 for host protection, i.e., via binding to host-derived mucins on the one hand and recognition and killing of bacteria on the other hand. The manuscript is well written and of interest to a wide readership.

The authors have convincingly addressed my concerns, either by providing additional data and/or discussion or by a reasonable rebuttal of the points that I raised. Thus, this already strong manuscript has been further strengthened during revision.

Reviewer #2

(Remarks to the Author)

All my questions have been answered clearly. I have no other points to raise.

Reviewer #3

(Remarks to the Author)

The authors have done a good job of responding to our questions and concerns. I look forward to seeing this exciting story published!

We thank the reviewers for their thoughtful, constructive comments, which have improved our manuscript. Our response to each comment is provided below.

Reviewer #1 (Remarks to the Author):

In their manuscript entitled "Intelectin-2 is a broad-spectrum antimicrobial lectin", Dugan et al. report an in-depth functional characterization of murine and human intelectin-2 (Itln2). The authors provide evidence for an interaction of both mItln2 and hItln2 with different host mucin isoforms as well as a variety of gram-negative and gram-positive bacteria, thus suggesting a role for Itln2 in microbial recognition. Moreover, they convincingly show direct antimicrobial effects of Itln2, thereby indicating a relevance for Itln2 in mucosal defense. In summary, the manuscript presents novel and important mechanistic insights into the role of intelectins in host-microbe interactions.

Overall, the present manuscript contains a wealth of experimental data that allow for elucidating the glycan preferences of both mItln2 and hItln2. It builds on previous studies that reported an upregulated expression of mouse Itln2 during nematode infection and postulated a role for Itln2 in inflammatory processes. By using an innovative combination of biophysical techniques, glycan arrays, biochemistry, and microbiology methods, the authors demonstrate a dual role of Itln2 for host protection, i.e., via binding to host-derived mucins on the one hand and recognition and killing of bacteria on the other hand. The manuscript is well written and of interest to a wide readership.

However, specific points need to be addressed to justify and/or support some conclusions drawn by the authors (see "Specific points" below).

Specific points:

1. Figure 1: Using enteroids from a congenic C57BL/6 mouse model encoding the complete, Itln1-6 locus, the authors show an induced Itln2 expression upon IL4 and IL-13 stimulation according to previous studies reporting Itln2 induction during Th2 responses. In this assay, Th1 cytokines and Th1-inducing inflammatory stimuli should be included to assess the specificity of the Itln2 upregulation by Th2 cytokines. Moreover, it will be interesting to see whether Itln2 is also induced upon stimulation of the enteroids with killed bacteria and/or bacteria-derived PAMPs.

Response: We appreciate the reviewer's request for more information on this point. In small intestinal tissue of conventionally housed/colonized mice, Itln2 transcript levels paralleled unstimulated (antibiotic-treated) enteroid cultures, indicating in the absence of stimuli, Itln2 expression is low. These data highlight the differences between the Itln2 and Itln1, which is constitutively expressed. Type1-associated cytokines, including interferon gamma, are reported to affect the viability and functions of Goblet and Paneth cells (<https://doi.org/10.1084/jem.20130753>; <https://doi.org/10.1172/jci.insight.121886>). This information and the cited reports have been added to the Results section (page 2). We found that type 2 cytokines, IL-4 and IL-13 selectively induce mItln2. As a control, the type 17/22 inflammatory cytokine, IL-22, had little or no effect on mItln2 expression. The effects of stimulating enteroids with microbes or microbe-derived PAMPs is an interesting research question that would require extensive optimization of experimental conditions, making it an intriguing avenue for future research.

2. Figures 2 and 5; Supplementary Tables 2 and 3: The authors determine the glycan specificity of mItln2 and hItln2 using different glycan arrays. They observe some hits for microbial glycans as well as mammalian glycans. However, the glycan array results require further explanation. For instance, it looks as if the RayBiotech mammalian glycan array yielded several (potentially unspecific?) hits, which seems to be in partial contrast to the CFG mammalian glycan array. Moreover, the fluorescence intensities are quite low in some instances (e.g., Figure 5E). It will be helpful and facilitate readability if the authors discuss the sometimes counterintuitive glycan array results in more detail and give further explanations.

Response: The reviewer helpfully highlights the need for a clearer explanation of our glycan array data. We revised the Results and Methods sections to address these concerns as follows:

- 1. Array differences:** The CFG and RayBiotech mammalian glycan arrays contain different glycan epitopes and utilize different surface chemistries, which contribute to the observed differences in binding patterns. The RayBiotech array includes some glycans likely to represent non-physiological epitopes. We added a discussion of these differences in the main text (page 3).
- 2. Signal intensities:** In running many glycan arrays, we and others have observed differences in fluorescence intensities between experiments. These variations in intensity arise from the differences in the glycan array slides and fluorescence scanners used. The mItln2 CFG mammalian glycan array data (Figure 2B) was collected in 2016 using ProScanArray Scanner (Perkin Elmer) instrument at the core facility. The microbial glycan array data (Figure 2B) and the RayBiotech mammalian array data (previously Figure S2B, now Figure S3B) for mItln2 was collected in 2021 in-house using Genepix 4400A scanner (Molecular Devices). Similarly, the hItln2 microbial glycan array data (Figure 5E) was collected in 2024 using a Genepix 4400A scanner (Molecular Devices) instrument at the core facility. During the 2024 run, the core facility noted that the scanner in use during that time had a maximum RFU of approximately 4,000 compared to 60,000 on previous scanner, resulting in lower Relative Fluorescence Unit (RFU) values. Moreover, in 2024, the core facility also changed protocols and began regenerating and reusing slides; the long-term effects on fluorescence intensity are unclear (<https://doi.org/10.1093/glycob/cwad091>). Despite these intensity differences, the binding patterns remain consistent within each experiment and were validated with biolayer interferometry. We added this information to the Methods section (page 15).
- 3. Data interpretation:** We also added a more detailed discussion of the glycan array results (page 3). We focused on the highest-intensity and the most reproducible hits. These align with our biochemical validation experiments.

3. Figures 2E, 5G (mucin binding of mItln2 and hItln2): The authors show a direct binding of mucins to mItln2 and hItln2 and convincingly confirm these interactions by different methods, including dot blot, fluorescence microscopy, and spectroscopic assays. However, how specific are the interactions of mItln2 with the different mucin isoforms? Are there any mucin isoforms that are not recognized by mItln2? Can the authors provide an explanation for the differential binding of hItln2 (e.g., why MUC5B is not bound by hItln2)?

Response: We appreciate the reviewer's thoughtful question regarding mucin specificity. In our binding assays, we found that mItln2 binds all major secreted mucins (MUC2, MUC5AC, MUC5B) relevant to intestinal and respiratory tract, as well as porcine gastric mucus (mixture of different mucins). However, hItln2's binding is more restricted, with strong affinity for MUC2 and MUC5AC. This differential binding may reflect subtle differences in j patterns (density, modification, and accessibility) between mucin isoforms. Given that both intelectins bind β -D-galactopyranose, differences in galactose accessibility or the influence of adjacent glycans likely contribute to selectivity. For example, porcine MUC2 displays glycans that are reported to be shorter and less extended than in MUC5AC and MUC5B (<https://doi.org/10.1038/s41522-023-00378-4>). We have expanded this observation and explanation in the Results section (page 7).

4. Supplementary Figure 6: To optimize hItln2 expression, the authors mutated some amino acid residues for an improved solubility and reduced aggregation. Could these modifications have affected the binding properties of hItln2?

Response: We appreciate the reviewer raising this issue. We introduced conservative changes in hItln2 to reduce aggregation without changing carbohydrate binding. These mutations were guided by structural homology models with hItln1 (>90% sequence similarity), where we mapped the hItln2 ligand binding site to those determined in the hItln1's crystal structure. We now provide more description for the changes we introduced (detailed in Figure S7C):

- 1. Installation of a glycosylation site in the N-terminal domain that is conserved in hItln1 and mItln2.** The specificities of these lectins are different though both have a glycosylation site, indicating glycosylation

does not have a role in binding. Introduction of an N-glycan in hItln2 improved protein expression and yields.

2. Removal of an unpaired cysteine (Cys 311) on a loop in the carbohydrate recognition domain distal from the ligand binding site. Replacing this cysteine reduced the aggregation of hItln2 protein, but the resulting protein had glycan-binding activity.

Independent evidence that the binding we detect is relevant is from a recent report on human lectin arrays, released while this manuscript was in submission. An Itln2 engineered to be monomeric (all cysteines involved in interchain disulfide linkages and unpaired Cys 311 were replaced) to afford a mutant protein that binds galactose (<https://doi.org/10.1016/j.jbc.2024.107869>).

The revised manuscript includes all the above information in the Results section (page 6).

5. According to the Methods section, binding studies to microbial isolates were done with live bacteria, while lectin binding to pathogenic isolates was determined using formaldehyde-fixed bacteria. Did the fixation procedure affect the binding behavior of the bacteria (did the authors compare Itln2 binding to live vs. fixed bacteria)?

Response: Previous studies have compared lectin binding to live versus formaldehyde-fixed bacteria and found that fixation had minimal impact on lectin binding patterns, suggesting that the surface glycans are preserved during fixation (<https://doi.org/10.1074/jbc.RA120.012783>). Biosafety limitations preclude running live pathogens at our flow cytometry core facility to compare all directly. We observe binding, however, to fixed pathogenic isolates and dose-dependent effects on live pathogenic isolates, including *S. aureus* and *K. pneumoniae*, in our growth assay (done in BL2 confinement). These data, and the cited literature, indicate that formaldehyde fixation neither impairs nor enhances lectin binding. The revised manuscript includes this information in the Methods section (page 16).

6. Titration of Itln2 for determination of antimicrobial activity: In some cases, the authors used different Itln2 concentrations for bacteria binding and killing assays, which renders comparisons of antimicrobial activities of one Itln2 against different bacteria species and/or among mItln2 and hItln2 difficult. This should be made consistent by titration of both lectins over a wider concentration range. For instance, in Figure 4B, it looks as if only the highest concentration of 5 μM mItln2 is active against *L. reuteri* (i.e. no activity up to 2.5 μM), whereas in a different assay for microbicidal activity, even the lowest concentration of 1 μM Itln2 exhibited a maximum antimicrobial effect. Thus, K_i values should be measured to determine the effective antimicrobial concentrations of mItln2 and hItln2.

Response: We appreciate the opportunity to clarify our experimental approach. The flow-based assays can identify microbial "binders" but are not quantitative measures of cell killing. Using flow cytometry, we observed that some, but not all, microbes showed changes in FSC vs SSC upon mItln2 treatment at 0.5 μM . We subsequently evaluated a panel of binders and found similar dose-dependent phenotypes in these isolates (Figure 4B and Figure S5, as the reviewer noted). The flow studies revealed concentrations that induced changes in cell size and scattering properties; they do not effectively measure antimicrobial activity. We employed CFU assays and growth inhibition studies for quantitative assessment of antimicrobial effects. In Figures 4E, 4G, and 4H, we found that lower concentrations of mItln2 (0.5-1 μM) affect the growth of various isolates, including *L. reuteri* and *K. pneumoniae*. These data indicate that mItln2 treatment exerts antimicrobial effects at lower concentrations.

We appreciate the value of standardized MICs, but direct comparison of lectin-mediated antimicrobial activity is difficult. Unlike traditional antibiotics that target specific cellular processes, lectins exert antimicrobial effects through glycan-binding interactions that can vary significantly between bacterial species due to differences in surface glycan sequences, cell wall composition, and glycan accessibility. As established in the antimicrobial susceptibility literature, MIC values are inherently specific to particular "drug/bug" combinations and can vary even between strains of the same species. The observed variation in Itln2 effectiveness between microbial isolates is expected and biologically meaningful. To address the reviewer's

concern, we have replotted the *K. pneumoniae* data for both hItln2 and mItln2 to establish MIC values for the same microbe using identical assay conditions for comparison (Figures S5M and S8J).

7. The authors found that hItln2 retained binding to beta-galactopyranose and additional ligands at low pH and ionic strength (Figure 7). Were similar experiments performed with mItln2 as well? EDTA addition did not abrogate the interaction of hItln2 with mucins and microbes at low pH and low salt concentrations, which is an interesting but somewhat unexpected finding. The authors suggest an “alternate binding mode” (p.7) as a potential explanation. Does that mean that the orientation of the carbohydrate ligands in their interaction with Itln2 is expected to be completely different under these conditions? Does such an altered binding mode have an impact on the specificity of the recognized microbes (i.e., is bacterial recognition of hItln2 altered, as compared with Supplementary Figure 7)?

Response: We did not test mItln2 at low pH and ionic strength because of its potent antimicrobial effects in neutral pH and physiological ionic strength at concentrations consistent with other antimicrobial proteins in the literature (0.5-5 μ M). Moreover, the isoelectric point of mItln2 is \sim 5.5, so low pH is likely to lead to protein aggregation.

We agree that mucin and microbial recognition by hItln2 under altered conditions is a fascinating observation and understanding its basis is a goal of future work. In the current study, we observed that hItln2 retained binding to “mucin and microbial binders” (Figures 7D, S8C, and S8E) and “galactopyranose and 6SO3LacNAc over galactofuranose) at low pH and ionic strength. For mucin, there was still a stronger interaction of hItln2 with MUC2 at low pH and ionic strength, with some weaker interaction with MUC5B and MUC5AC. Moreover, treatment of human fecal samples with hItln2 at both conditions yielded similar percentage of bound microbes (Figures 6A and 7B), therefore, the specificity of hItln2 for the recognized mucins and microbes is largely retained. Further analyses, using methods such as lectin-seq (<https://doi.org/10.1126/sciadv.add8766>) and glycan array studies of hItln2 under low pH and ionic strength conditions, are major undertakings that could shed further light into whether the specificity of hItln2 is expanded in such conditions.

Reviewer #2 (Remarks to the Author):

This is a fine manuscript describing an animal lectin for which no activity could be assigned previously. The extensive study brings molecular characterization, specificity study with some measurement of affinity, and functional role against pathogen infection. The manuscript reads very well, but, presumably because of space limitation, lacks some better description of the large amount of experimental data.

Major points

1. The oligomeric state of the proteins is quickly discussed and some more description of experimental data would be helpful. If mItln2 is composed of a mixture of different oligomers, does this mean that only part of the sample is correctly associated? what about the effect on activity? As for hItln2, it is stated that it occurs as a covalent trimer, but supp. Fig 6e is more in evidence of a hexamer (Supp Fig 6d should be explicated). A Cys involved in oligomerization has been mutated, what is the effect on oligomerization?

Response: We thank the reviewer for highlighting this aspect. We now provide a more detailed characterization of the oligomeric states of the lectins in the Results sections (pages 3, 6, and 7). MItln2 exists as a mixture of dimers, trimers, and higher-order oligomers as determined by DLS and crosslinking studies (Figures S2A and S2B). Attempts to isolate specific oligomerization states by size exclusion chromatography were unsuccessful because the lectin typically bound to standard matrices (Sephacrose). Moreover, literature precedents for other lectins suggest that oligomerization is driven by multivalent glycan ligands and that multiple oligomeric forms can retain activity (<https://doi.org/10.1146/annurev-biochem-062917-012322>). For example, MBL (a disulfide-linked trimer) can form oligomers up to dodecamers that all bind mannose and recruit complement (<https://doi.org/10.1016/j.imbio.2008.11.003>). Similarly, RegIIIgamma functions can bind oligomerize upon

binding its membrane ligands to form hexameric pores (<https://doi.org/10.1038/nature12729>). These data suggest that mItln2 oligomers have functional activity, an interesting avenue for future studies.

The reviewer notes that Figure S6E (now Figure S7F) shows that hItln2 can form trimers and hexamers. The N-terminal cysteines involved in hItln1 oligomerization are conserved in hItln2 and preserved in our study. We observe stable covalent oligomers through intermolecular disulfide bonds. hItln2 has an additional unpaired cysteine near the C-terminus that led to aggregation issues, but the location of this cysteine is inconsistent with a role in physiological oligomerization. Moreover, our observation of recombinant hItln2 forming trimers and hexamers is consistent with the oligomeric state reported for native hItln2 in human ileal tissue (<https://doi.org/10.1096/fj.202101870R>). That we observe the same oligomerization behavior as native protein is reassuring evidence that our recombinant protein behaves like the endogenous protein. This information is described in our revised manuscript. Similarly, the DLS data for hItln2 (previously Figure S6D, now Figure S7E) shows hydrodynamic radius value of around 40 nm, suggesting the presence of higher order oligomers. These data provide a qualitative measure of the oligomeric state, as DLS analysis assumes that the protein is globular, is more biased for larger particles; thus, it cannot determine the oligomeric state of the protein. So, our estimation of the oligomeric states of hItln2 was mainly guided by the SDS-PAGE data. Lastly, we point to a recent publication that emerged as we were submitting this manuscript suggesting that monomeric hItln2 can bind (<https://doi.org/10.1074/jbc.RA120.012783>). Together, the data indicate that oligomerization is beneficial for avidity but not required for binding.

2. The glycan array data are of high importance. They provide the basis of the lectin specificity that was unknown for intelectin2. Fig 2B should include identification (labeling) of the highly positive ligands. Some more discussion about the observation would be of interest. In the mammalian array, results pointing to N-glycans and to O-glycans should be discussed: biantennary glycan with LacNAc repeats are excellent ligands, specially those with 3 LacNAc repeats (540 and 567) and no binding is observed with only 2 repeats. Triantennary is even better. More numerous LacNAc repeats result in weaker binding. This could be discussed in term of lectin topology. Similarly, the binding of terminal sulfated Gal is not that simple: it is efficient only when bound to 6SGlcNAc or 6SGlc by b1-4 linkage. Figure 2C is rather misleading: the two disaccharides at the bottom are not recognized by the lectin.

Response: We appreciate the reviewer's thoughtful advice regarding the analysis of the glycan array data. We added labels to identify the highest-binding ligands in Figure 2B. Additionally, we expanded our discussion on the structure-activity relationships (pages 3 and 4). We agree with the reviewer's observations about biantennary and triantennary N-glycans with LacNAc repeats being excellent ligands for mItln2 and have included comments about lectin topology as possible determining factor influencing the interactions. We have corrected the disaccharide structure in Figure 2C to reflect the glycan hits from the CFG microbial and mammalian arrays. However, we clarify that the two disaccharides [LacNAc disaccharide and 6SO₃LacNAc (with sulfation on galactose but not glucose)], listed previously at the bottom in the Figure 2C are recognized by mItln2 as demonstrated by the ELLA and biolayer interferometry assays (Figures S3G-S2J). Additionally, an important limitation in interpreting glycan array data is the lack of information on relative affinities or ranked binding preferences. The apparent differences in signal intensity may reflect differences in array density, presentation context, or other technical factors rather than true affinity differences. We therefore validated the glycan array results through complementary biochemical approaches. We incorporated this more nuanced discussion of the glycan array results and their interpretation limitations into the main text, while emphasizing that these data provide the foundation for our subsequent detailed characterization of mItln2 binding specificity (page 3).

3. It is not clear why hItln2 was not assayed on the mammalian array.

Response: The hItln2 protein was tested on both mammalian and microbial glycan arrays by the CFG, but this run appeared to fail as it had no binding on the former. We postulate there was a problem with the array or that the protein was no longer active, as we have encountered this issue previously with different proteins. We since tested hItln2 on select mammalian glycans using alternative approaches (dot blot, biolayer interferometry, and

microscopy) and confirmed hItln2 and mItln2 have similar binding patterns. We added this clarification to the Methods section (page 15).

4. The excel files provided in supplemental data are not easy to understand (except for CFG mammalian). There is no identification of the polysaccharides/microbes in the microbial analysis, and only code names for the RayBiotech data. This glycan array is never mentioned in the main text (Supp Fig 2b is cited with no explanation) and the strong discrepancy between SFG glycan array and RayBiotech is not discussed.

Response: We agree that presenting more of the glycan data would be helpful. We added comprehensive legends and identifiers to all supplemental data files. We additionally included references for glycan structure identifications for the microbial array data. We have also amended the main text to discuss the RayBiotech array results in the main text, as noted in our response to Reviewer #1 (page 3).

5. 3D-models are important for comparing intelectin-2 (mouse and human) with intelectin-1. In the present version, one model is obtained with Swissmodel and the other with Alpha-Fold3 without justification. The same procedure (either one or the other) should be used for both. The two methods also allows for modeling oligomeric state and it would be of interest to have a view of the modelled trimers. Alignment of sequences is provided only for 2 sequences in Supp Fig1a. It would be useful to have a global alignment for mouse and human intelectins (1 and 2). Finally docking galactose would not be much work (for example with Autodock) and will rationalize the effect of sulfate (but of course could be kept for further study).

Response: We thank the reviewer for these suggestions. We have now generated all structural models using AlphaFold3 for consistency and have included predicted oligomeric assemblies for both proteins (Figures S2D and S7G). We have also included alignment of mouse and human Itln1 and Itln2 (Figure S1B). We agree that docking studies would be informative; however, the current models are not especially adept at modeling calcium-dependent interactions. Structural studies are underway with these lectins, and we hope to report a binding mode in our future studies.

Reviewer #3 (Remarks to the Author):

This manuscript provides important new insights into mucosal immunity by characterizing the intelectin-2 (Itln2) family in both mice and humans. Building on the authors' prior work on intelectin-1 (Itln1), the study reveals distinct and complementary roles for Itln2 in host defense at mucosal surfaces. These lectins are expressed by epithelial cells in barrier tissues, and the current study focuses on their function in intestinal immunity.

The authors show that mouse Itln2 expression is induced by type 2 cytokines and that the protein exhibits antimicrobial activity via direct bacterial killing and membrane disruption. In contrast, human ITLN2 primarily agglutinates bacteria. Both forms can cross-link mucins, suggesting a dual role in antimicrobial defense and enhancing mucosal barrier integrity. Importantly, the authors demonstrate that mItln2 binds terminal β -D-galactopyranose, a specificity distinct from mItln1 and hITLN1. This divergence is supported by structural data.

Overall, the data are quite robust and the conclusions are well-supported. The work makes a significant contribution to our understanding of the diversity of antimicrobial proteins (AMPs) at epithelial surfaces. That said, I have a few specific questions and suggestions that could further strengthen this already strong manuscript:

1. The microarray data indicate that intelectins also bind to mammalian glycans. More discussion is warranted on how carbohydrate structure enables discrimination between host and microbial targets. What prevents host cell targeting in vivo?

Response: Several factors could contribute to host vs. microbial discrimination by intelectin-2. Firstly, Itln2 is secreted by Paneth and goblet cells into the mucus layer where it encounters high concentrations of secreted mucins (which it cross-links) and resident or invading microbes. This locale gives Itln2 limited access to host

epithelial cell surfaces. Additionally, microbial surface glycans may present β -galactopyranose residues at a higher density, thereby facilitating preferential recognition of intelectin-2 to microbes. We expanded the discussion to delineate these possibilities (page 9).

2. Several known AMPs show preferential activity against log-phase bacteria. Has the antimicrobial activity of mouse Itln2 been tested across different bacterial growth phases?

Response: Our initial assays for Itln2 binding and activity against microbial isolates were conducted in log and stationary phases. We saw binding and antimicrobial effects in both growth phases. Given these results, we performed assays at mid-log as this is the most standard format in the literature, and it allowed us to use a consistent protocol between different microbial isolates.

3. The liposome disruption assays use mammalian-type lipid compositions, yet these membranes are disrupted by mItln2. Can the authors clarify why they think that these proteins don't show toxicity to mammalian cell membranes?

Response: We thank the reviewer for this clarifying question. This apparent paradox likely reflects several factors, including the complexity of cell membranes compared to liposomes. Our liposomes contain simplified lipid compositions, while mammalian cell membranes have complex asymmetric lipid distributions, variable cholesterol content, and associated proteins that may resist disruption. We additionally postulate that while ligands present on mammalian cells may result in Itln2 binding, the differences in the location and distribution of mammalian glycans preclude Itln2 assemblies that lead to cell killing. For example, the Reg proteins are believed to bind abundant ligands on microbial cells and then assemble into hexameric pores that result in cell lysis (<https://doi.org/10.1038/nature12729>). If Itln2 operates through a similar mechanism, the lectin likely binds to the most accessible galactose residues of the glycocalyx. In mammalian cells, these residues may be too far away from the membrane to assemble into a pore. Moreover, Itln2 is secreted in environments rich with microbes and mucins, and these components may compete for binding or create local concentration gradients that favor microbial targeting over mammalian cell interaction. We have incorporated these discussions in the main text (page 9).

4. What is the estimated concentration of mouse Itln2 in the intestine or enteroids? This would help contextualize the in vitro findings.

Response: There are no accurate measures of mItln2 concentration in the intestine or enteroids. Proteomic analyses of the enteroids and mouse tissue would be informative for future work examining mItln2 function in vivo. Still, mItln2 has been reported to be one of the most abundant proteins in the BALB/c mouse jejunum following nematode infections (<https://doi.org/10.1002/pmic.200300658>). Moreover, in murine organoid cultures, mItln1 transcripts are expressed abundantly at levels comparable to those found in tissue (<https://doi.org/10.1073/pnas.2312453120>), suggesting a correlation between enteroid and tissue expression. Thus, we reason that mItln2 is abundantly expressed in the intestine and enteroids in the type-2 inflammatory conditions. Moreover, hItln2 is even more highly expressed than hItln1 in ileal tissue. Association of Itln2 with the mucus layer would result in very high local concentrations that would be difficult to assess by analyzing soluble lectin alone.

5. The section on pH sensitivity could be clarified. While the authors note that low pH and low salt are linked to inflammation, gut epithelial surfaces also experience these conditions physiologically. Where are Itln2 proteins expressed relative to such environments? Notably, their ability to function at pH 7.4 and 150 mM NaCl is unusual for AMPs and could be emphasized.

Response: We thank the reviewer for this comment. Under normal conditions, hItln2 is expressed by Paneth cells in the distal small intestine (highest expression is in the ileum). A healthy ileum, a site of nutrient

absorption, tends to be neutral pH and have physiological salt concentration. In contrast, mItln2 is not expressed in healthy mice but by mucus-secreting cells in the intestines and lungs during type 2 inflammation. We now elaborate on the distinct abilities of these lectins to function at physiological pH and ionic strengths in the Discussion sections (pages 8 and 9).

6. How do intelectins complement or differ from other AMPs? In other words, how are they solving a distinct “evolutionary challenge”? Is it just that they are induced by type 2 cytokines? Their broad bacterial specificity would suggest that they are not simply targeting a subset of bacteria that is “missed” by other AMPs.

Response: We thank the reviewer for this excellent question. Though more information on the microbial specificity of other antimicrobial lectins and AMPs is needed to answer this question fully, intelectins represent an evolutionarily ancient solution to pathogen recognition that predates the adaptive immune system. Their high conservation across chordates suggests they fill a need for immediate, broad-spectrum antimicrobial activity without requiring the clonal selection that characterizes adaptive immunity. Moreover, in contrast to canonical AMPs that directly disrupt bacterial membranes through electrostatic interactions, intelectins function through more specific targeting of distinct carbohydrate epitopes. Additionally, the family can participate in a range of effector functions, including immune cell recruitment, a property that also distinguishes them from classic AMPs. We found that Itln2 not only engages microbes but also mucins to establish different modes of defense. In this way, by altering the mucosal environment, it can protect against microbes for which direct recognition is lacking. This more passive defense mechanism is advantageous in clearing infections in tissues where inflammatory collateral damage needs to be reduced. We believe these properties set the intelectins apart from AMPs in the mucosal environment. We highlighted these properties in the Discussion (page 9).

7. Since type 2 cytokines promote goblet cell hyperplasia and mucus production, the co-induction of antimicrobial intelectins that also cross-link mucins is intriguing. This synergistic relationship could be highlighted more explicitly in the discussion.

Response: We appreciate the reviewer’s advice. The dual functions of Itln2 suggest it evolved to address a specific challenge in barrier immunity: the generation of antimicrobial barriers that are structurally robust and functionally active. This is particularly critical when tissues are undergoing active remodeling and repair during type 2 inflammation. We have taken the reviewer’s advice to augment the discussion of this issue (page 9).

8. Given the role of type 2 cytokines, are STAT6 binding motifs present in the promoters of the mItln2 and hITLN2 genes?

Response: We appreciate this reviewer suggestion. There is evidence that mItln2 is a STAT6-dependent gene (<https://doi.org/10.1016/j.exppara.2007.02.015>), which we now cite. Additionally, analysis of promoter sequences of mItln2 (Figure S1A) and hItln2 (Figure S7A) showed presence of STAT6 binding motifs. However, we note that the Th2-dependent upregulation of hItln2 (which is constitutively expressed) require further confirmation.

9. Do N-linked glycosylation sites on the recombinant proteins affect their oligomerization state or binding activity? Are any of the N-glycans capped with β -D-galactopyranose, potentially affecting specificity?

Response: mItln2 is naturally N-glycosylated and introducing an N-glycosylation site in hItln2 helped improve protein expression and solubility. If any Itln2 self-association occurs, it would lead to competition between the protein and other targets on the cell surface or mucins, but we observe robust binding to cells and mucins as well as immobilized glycans. The observation that glycosylation of hItln2 improves its solubility also supports that no competing self-interactions occur with the lectins, as these could lead to formation of large aggregates.

Reviewer #4 (Remarks to the Author):

We thank this reviewer for their transparency and valuable comments, which improved our manuscript.